# Next generation genetically encoded fluorescent sensors for serotonin

Martin Kubitschke[1], Monika Müller [2], Lutz Wallhorn [1], Mauro Pulin [3], Manuel Mittag[2], Stefan Pollok[4], Tim Ziebarth [4], Svenja Bremshey[1], Jill Gerdey [1], Kristin Carolin Claussen[1], Kim Renken[1], Juliana Groß[1], Pascal Gneiße[1], Niklas Meyer[1], J. Simon Wiegert [3,5], Andreas Reiner [4], Martin Fuhrmann [2] & Olivia Andrea Masseck [1] ✉

We developed a family of genetically encoded serotonin (5-HT) sensors (sDarken) on the basis of the native 5-HT1A receptor and circularly permuted GFP. sDarken 5-HT sensors are bright in the unbound state and diminish their fluorescence upon binding of 5-HT. Sensor variants with different affinities for serotonin were engineered to increase the versatility in imaging of serotonin dynamics. Experiments in vitro and in vivo showed the feasibility of imaging serotonin dynamics with high temporal and spatial resolution. As demonstrated here, the designed sensors show excellent membrane expression, have high specificity and a superior signal-to-noise ratio, detect the endogenous release of serotonin and are suitable for two-photon in vivo imaging.

Evidence has accumulated that serotonin (5-HT), a hormone and neurotransmitter, is not only involved in a variety of different physiological and central nervous system functions but also in the development and manifestation of psychiatric diseases, such as anxiety and depression. Although the monoamine hypothesis of depression was already postulated in the 1950s, the contribution of 5-HT to the development of these psychiatric diseases still needs to be elucidated in more detail. A reason for the lack of knowledge is that no tools were available to measure the release of 5-HT with high temporal and spatial resolution. Until recently, serotonin concentrations have primarily been measured by microdialysis with subsequent HPLC quantification —an invasive technique with low temporal and spatial resolution[1]. To circumvent these drawbacks, fast-scan cyclic voltammetry (FSCV) and amperometry are used as an alternative for measuring serotonin concentrations with high temporal resolution[2,3]. FSCV measures the redox current at the surface of a carbon-fiber electrode to draw conclusions about the oxidized or reduced analytes at the electrode tip. However, since several electrochemical active substances can be oxidized or reduced at overlapping potentials, FSCV for serotonin suffers from a lack of specificity[2,3].

In addition, with these techniques it is neither possible to monitor serotonin dynamics across larger areas nor to achieve single cell or even synaptic resolution[4-6]. As a neuromodulator, serotonin is not only acting via synaptic transmission in a point-to-point manner, but also via volume transmission. The direct relationship between serotonergic cell firing, serotonin release, synaptic vs. volume transmission, reuptake and basal serotonin levels are not well understood and will remain unanswered until serotonin release can be imaged in real time in the awake, behaving animal[7].

In the last years, genetically encoded sensors have been developed and became versatile tools for imaging neuronal activity, especially for imaging the release of different neurotransmitters, such as glutamate, dopamine, norepinephrine, acetylcholine and very recently serotonin[7-10]. Most of the genetically encoded sensors use a specific G-protein coupled receptor (GPCR) as sensing moiety fused to circularly permuted fluorescent protein. Another sensor design is based on periplasmic binding proteins (PBP) that have been developed to measure neurotransmitters such as glutamate, acetylcholine and γ-aminobutyric acid (GABA)[11,12]. Very recently a PBP-based 5-HT sensor (iSeroSnFR) was obtained by directed evolution of the acetylcholine

[1]Synthetic Biology, University of Bremen, Bremen, Germany. [2]Neuroimmunology and Imaging Group, German Center for Neurodegenerative Diseases (DZNE), Bonn, Germany. [3]Synaptic Wiring and Information Processing, Center for Molecular Neurobiology (ZMNH), UKE, Hamburg, Germany. [4]Cellular Neurobiology, Department of Biology and Biotechnology, Ruhr University Bochum, Bochum, Germany. [5]Department of Neurophysiology, University of Heidelberg, Mannheim, Mannheim, Germany. ✉e-mail: masseck@uni-bremen.de

sensor[13]. In this work we expand the existing toolbox to overcome constraints of available sensors.

## Results

We designed different variants of a genetically encoded serotonin sensor to image serotonin dynamics with high temporal and spatial resolution. We decided to use the native human 5-HT$_{1A}$ receptor (UniProt: P08908) as the sensing scaffold due to its high affinity to 5-HT, compared to other 5-HT receptor subtypes[14]. Our initial sensor design is based on the recently described genetically encoded dopamine sensor dLight1[8] (Fig. 1a). Sequence alignments of transmembrane domains of DRD1, 5-HT$_{2A}$ and 5-HT$_{1A}$ were performed to identify the optimal insertion site of cpGFP (Supplementary Fig. S1). In all variants, the third intracellular loop of the native 5-HT$_{1A}$ receptor was replaced by the circularly permuted form of GFP (cpGFP) also used in GCaMP6m[15], this means also mutations that were introduced during the engineering of GCaMP6m are included (AS 65, 75, 87, 92, 115, 118, 250). cpGFP flanked by LSSLE (linker1) and LPDQL (linker2) was inserted between F224 and R337 of the native 5-HT$_{1A}$ receptor (Supplementary Fig. S1). Starting from this initial variant we performed mutagenesis of the linker residues to change the properties of the 5-HT sensor. In a first optimization step, we randomly mutated the amino acids at positions 4 and 5 of linker 1 and the amino acids at position 1 and 2 of linker 2 (Supplementary Fig. S1). In total, we measured and analyzed a library of 224 mutants (for details see Supplementary Fig. S2) in human embryonic kidney (HEK) cells. Interestingly, most of the mutants except for two (M24, M27) diminished their fluorescence upon serotonin binding (Fig. 1b). Therefore, we decided to carry on with the optimization of a darkening sensor (i.e., reduction of fluorescence upon 5-HT binding).

Mutant M34 was the most promising candidate with excellent membrane trafficking properties as well as a robust decrease in fluorescence ($\Delta F/F_0 = -0.71 \pm 0.01$, mean ± SEM, $n = 15$) after the application of 800 nM 5-HT (Fig. 1b, c, Supplementary Movie 1). Based on the special feature of our serotonin sensor M34, namely its fluorescence decreases after serotonin binding, we named it sDarken (serotonin darkening 5-HT$_{1A}$ receptor-based sensor).

### In vitro characterization of sDarken

As in vivo applications require robust responses and photostability, we investigated sensor responses to repeated stimulation with 5-HT (Fig. 1d–f). sDarken could be activated repetitively (Fig. 1d–f, Supplementary Movie 2). We validated our findings by 5-HT uncaging experiments. Repetitive light-evoked release of 5-HT caused a decrease in fluorescence (Fig. 1e). Furthermore, ultrafast 5-HT application to outside-out patches of HEK cells expressing sDarken showed stable response amplitudes upon repeated applications (Fig. 1f).

As expected, sDarken had a high affinity to serotonin with a $K_d$ of $127 \pm 20.7$ nM (mean ± SEM, $n = 10$) (Fig. 1g) and sDarken was thus able to reliably detect serotonin concentrations between 100 nM and 1 μM. sDarken did not show any significant responses to physiological concentrations of structurally similar substances or other neurotransmitters (Fig. 1h). The application of serotonin was more effective than the application of the selective 5-HT$_{1A}$ agonist 8-OHDPAT, which induced a significantly lower decrease in sDarken fluorescence than application of serotonin (5-HT = 0.7 $-\Delta F/F_0 \pm 0.02$, 8-OHDPAT = 0.53 $-\Delta F/F_0 \pm 0.02$, $p < 0.001$). However, neither application of the serotonin precursor L-tryptophan nor its degradation product 5-hydroxyindoleacetic acid (5-HIAA) did elicit any substantial sensor responses (L-Tryp $-0.02 -\Delta F/F_0 \pm 0.01$, 5-HIAA $0.02 -\Delta F/F_0 \pm 0.02$). Next, we tested if other monoamines, such as dopamine, histamine and norepinephrine (NE) would result in a fluorescence decrease of sDarken. Neither the application of dopamine, histamine nor NE caused any changes in fluorescence (dopamine $0.03 -\Delta F/F_0 \pm 0.01$, histamine $-0.06 -\Delta F/F_0 \pm 0.05$, ΔE $-0.02 -\Delta F/F_0 \pm 0.02$). Also, the

inhibitory neurotransmitter GABA elicited no significant fluorescence changes (GABA $-0.02 -\Delta F/F_0 \pm 0.01$). Similarly, the most abundant excitatory neurotransmitter glutamate did not elicit any response (glutamate $0.00 -\Delta F/F_0 \pm 0.01$). The same holds true for acetylcholine, as it did not induce a decrease in fluorescence of sDarken (acetylcholine $-0.07 -\Delta F/F_0 \pm 0.01$). Responses to other neurotransmitters and similar substances were not significantly different from the application of PBS only ($0.02 -\Delta F/F_0 \pm 0.02$) (Fig. 1h).

In contrast to that, the application of the selective 5-HT1A antagonist WAY-100635 (WAY) blocked responses to 5-HT. As expected, the application of WAY also reversed the decrease in fluorescence elicited by an initial application of 5-HT and behaved like an inverse agonist (Supplementary Fig. S3). To assess whether sDarken is sensitive to pH change fluorescence of sDarken was evaluated in different pH buffers with and without the presence of serotonin (Supplementary Fig. S4). In addition, we analyzed fluorescence changes of sDarken in buffer solutions of different pH values (pH 6–8). Only minimal differences in responses of sDarken were observed to pH changes in the range of pH 6–8. No significant differences in fluorescence of sDarken were observed to pH changes in the range of pH 6.8 –7.4. In addition, dose response curves at different pH values show that affinity and dynamic range is not substantially effected by pH changes (Supplementary Fig. S4).

In summary, sDarken showed robust and large signal changes to submicromolar 5-HT concentrations and no detectable responses to other neurotransmitters and similar substances, in vitro.

### Intracellular signaling pathways

Due to the substitution of the third intracellular loop of the 5-HT$_{1A}$ receptor by cpGFP, the native coupling and activation of the G proteins should be disrupted[8]. Indeed, we found that $\beta\gamma$-mediated opening of G protein-coupled inwardly rectifying potassium channels (GIRK) was abolished (Supplementary Fig. S5). Also, other G protein-coupled pathways (i.e., $G_q$-signaling and $G_s$-signaling) showed no activation upon binding of serotonin (Supplementary Figs. S6 and S7). Taken together, the exchange of the third intracellular loop with cpGFP disrupts G protein binding and thus sDarken does thus not trigger the activation of endogenous signaling pathways.

### Dynamic range and affinity

Depending on the research question, in vivo applications will require different 5-HT sensors with varying affinities to 5-HT: For detecting volume transmission it would be beneficial to utilize a sensor with a very high affinity to 5-HT, while for the detection of high frequency burst of 5-HT release at serotonergic synapses, it would be favorable to have a sensor with a lower affinity. We, therefore, investigated whether mutations within the binding pocket of sDarken can be used to modify its affinity to 5-HT. We performed site-directed substitutions of conserved aspartate and serine residues within the serotonin binding pocket to obtain low affinity versions of sDarken[16]. D82N, D116N, S198A were introduced and investigated in all possible combinations. All mutations had a lower affinity for serotonin and D116N seemed particularly promising. As already seen for the initial sDarken-sensor design, D116N expression was restricted to the membrane (Fig. 2a), while titration curves revealed a 1000-fold lower affinity for serotonin ($K_d$ $45 \pm 4.6$ μM) in comparison to sDarken ($K_d$ $127 \pm 20.7$ nM) (Fig. 2c and Supplementary Movies 3–5). The detection range of D116N variant lies between 100 μM to 1 mM (Supplementary Movies 3–5) and the variant therefore is named low affinity sDarken (L-sDarken).

To obtain an increased sensitivity for serotonin we decided to exchange cpGFP with the superfolder version of cpGFP[17]. The advantage of superfolder cpGFP is its superb ability to fold correctly even when fused to proteins. In different biosensors, i.e., GCaMP, $Zn^{2+}$ and voltage sensor) an exchange of cpGFP with the superfolder variant improved sensitivity, dynamic range, brightness and photostability[17,18].

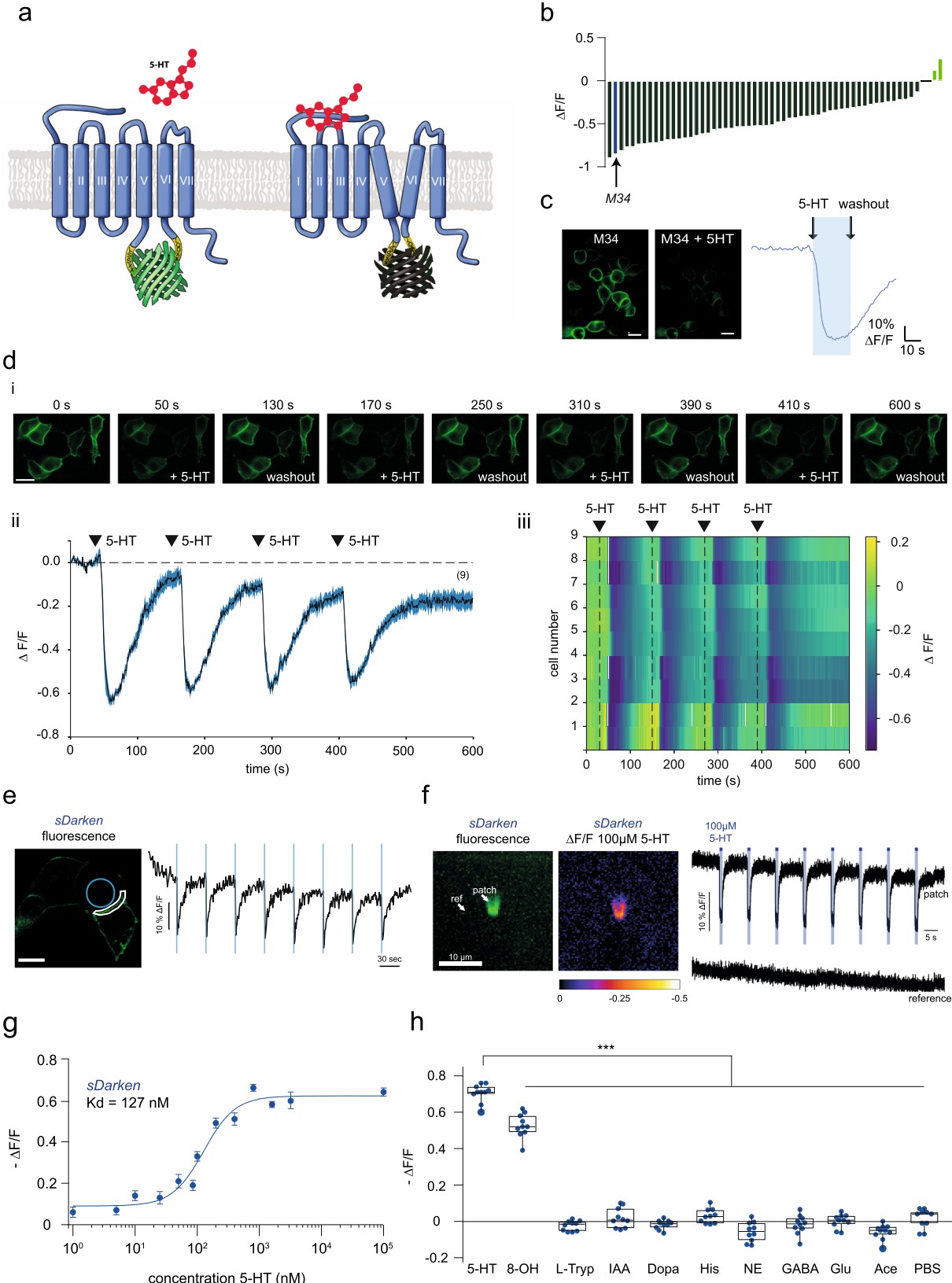

Hence we hypothesized that using a superfolder variant of cpGFP would increase the initial fluorescence and potentially decrease $K_d$ values[17,18]. Replacement of cpGFP with the superfolder cpGFP improved sensitivity and yielded a bright and photostable sensor that is well expressed in the membrane. Similar to sDarken and L-sDarken, superfolder sDarken decreases its fluorescence in response to 5-HT

application (Fig. 2b and Supplementary Movies 6 and 7). Indeed, substitution of cpGFP with the superfolder cpGFP increased the affinity twofold to 57 nM ($57.3 \pm 13.9$ nM) in comparison to the original sDarken (Fig. 2c), whereas the signal amplitude remained unchanged (Fig. 2c). From now on we name this high affinity variant H-sDarken. Similar to the original sDarken, no substantial responses to other

**Fig. 1 | Characterization in HEK cells. a** Design principle of *sDarken*. **b** Screening result of linker variants. Bars indicate fluorescence changes upon application of serotonin. Blue bar represents mutant M34. **c** Expression of mutant M34 (*sDarken*) in HEK cells. Scale bar 20 µm. Fluorescence before and after the application of 800 nM 5-HT. Representative fluorescence trace of sensor variant M34 expressing HEK cell during the application of 800 nM 5-HT. Experiments were at least repeated three times. **d** (i) Representative image sequence of repetitive 5-HT application via wash in. (ii) Example fluorescence measurement, repetitive application of 5 µM 5-HT in 9 example cells in two dishes. Application of 5-HT leads to a reversible reduction of fluorescence. No baseline corrections were performed. (iii) Heatmap of all analyzed ROIs. Scale bar in the first frame applies to all following frames. Scale bar 20 µm. **e** Example trace of 5-HT uncaging (405 nm, laser power: 90%, bleaching period: 90 ms, interval: 50 s) evoked *sDarken* responses. Blue circle indicates uncaging) area. White line indicates the part of the membrane that was used for analysis. Fluorescence signal of sDarken. Δ*F/F* values of part of membrane (white

line) over time (exemplary trial). Blue bars indicate uncaging (i.e., bleaching) intervals. Repetitive uncaging of 5-HT resulted in decreasing ΔF/F values of sDarken. **f** Fast 5-HT application to an outside-out patch. Representative image from an outside-out patch containing sDarken 48 h post transfection (left). Fluorescence intensity changes in the patch. The fluorescence recovery after 5-HT wash out is significantly slower than the activation. No background/baseline corrections were performed. **g** Dose response curve measured in response to different 5-HT concentrations. Group data *n* = 20 from at least 2 replicates, mean and ±SEM are shown. **h** Fluorescence changes after application of serotonin, related substances or neurotransmitters (if not mentioned differently, concentration 10 µM (*n* = 10). Box represents the 25% percentile to the 75% percentile. The line in the middle of the box represents the median. One-Way ANOVA Multiple Comparison, ***$p < 0.001$, 5-HT serotonin, 8-OH 8-OHDPAT, L-Tryp tryptophan, IAA 5-hydroxyindoleacetic acid, Dopa dopamine, His histidine, NE norepinephrine, GABA Glu glutamate, Ace acetylcholine, PBS phosphate buffer.

neurotransmitters or substances were evident neither for *L*-sDarken nor for H-sDarken (Fig. 2d and Supplementary Fig. S8). Also, no activation of endogenous intracellular signaling pathways could be observed for L-sDarken and H-sDarken (Supplementary Figs. S5–S9).

## Photostability and membrane expression

Since sDarken is a genetically encoded sensor which decreases its fluorescence upon binding of serotonin, photostability during recordings is a prerequisite for long-term imaging. In comparison to the membrane-targeted expression of eGFP (CAAX-motif), all three sensor variants showed similar or even increased photostability (Fig. 2e). In general, sDarken has a high baseline fluorescence in the unbound state, making it easily detectable against background and autofluorescence (Supplementary Fig. S10a). Next, we analyzed membrane trafficking and expression of sDarken. No significant difference between membrane-targeted expression of eGFP and 5-HT sensor variants was evident (Supplementary Fig. S10b). All three variants expressed equally well in the membrane and did not show any noteworthy intracellular aggregation (Fig. 2f). In addition, we wanted to know if long-term exposure to serotonin might induce permanent internalization of our sensor, which proved not to be the case (Supplementary Fig. S11).

Next, we validated the expression and performance of sDarken and its variants H-sDarken and L-sDarken in mammalian hippocampal neurons with two-photon microscopy. We individually expressed the three sDarken variants together with a cytosolic red-fluorescent protein, tdimer2, by single-cell electroporation in CA1 pyramidal neurons of organotypic hippocampal slice cultures. A few days after transfection, all three sensor variants showed strong, membrane-localized expression along the somatodendritic compartment (Fig. 2g, h). This expression profile suggests that the three sDarken variants can be used as reporters of synaptic point-to-point serotonin release events at axons, but also postsynaptically at dendrites and spines. Given the fact that sDarken is a negatively responding sensor and therefore bright in the non-ligand bound state, we evaluated its photostability under continuous two-photon excitation for 1 min. All three sDarken variants displayed a stable fluorescent signal throughout the imaging session (99.99% of baseline for sDarken, 99.98% for L-sDarken and 99.98% for H-sDarken) unlike tdimer2, which showed mild bleaching under these conditions (Fig. 2h and Supplementary Movies 8 and 9).

## Sensor kinetics

We next characterized the temporal characteristics of the 5-HT sensor variants. To achieve fast ligand application and removal, we used membrane patch fluorometry in conjunction with rapid, piezo-driven solution exchange[19]. Outside-out membrane patches from sDarken-expressing HEK cells were excised with a patch pipette and placed in front of a piezo-mounted double-barreled theta-glass pipette, which allows for submillisecond switching between a solution containing

5-HT and solution without 5-HT (Supplementary Fig. S12 and Supplementary Methods). The sensor showed a bright and stable fluorescence signal in the dome region of the patch (Fig. 1f and Supplementary Movie 10).

Repeated application of 5-HT resulted in characteristic signal responses of sDarken (Fig. 1e, f). High concentrations (100 µM 5-HT) caused a fast and pronounced drop in fluorescence intensity (ON) followed by a slower recovery of the fluorescence intensity after 5-HT removal (OFF) (Fig. 3a, b and Supplementary Movie 10). The stability of the patch, the high photostability of the sensor, and the high signal-to-noise ratio allowed for repeated measurements over extended times (Fig. 1f). The signal decreases Δ$F/F_0$ at individual patches ranged between 0.26 and 0.62 (0.43 ± 0.13, mean ± sd, *n* = 9 patches; Fig. 3a). The maximal signal change is in good agreement with data from whole cells (100 µM 5-HT, Δ$F/F_0$ = −0.64 ± 0.02), whereas smaller changes seen in some patches can be likely attributed to sensor populations that are not exposed to the solution but are in contact with the glass surface of the patch pipette. Application of extracellular solution without 5-HT did not elicit any signal changes (*n* = 6 patches; Fig. 3a).

The activation and deactivation kinetics of sDarken were fitted single exponentially (see "Methods" for details). For application of 100 µM 5-HT we obtained $\tau_{ON}$ = 43.5 ± 9.7 ms, which approaches the time resolution of imaging (91 fps), while deactivation was considerably slower ($\tau_{OFF}$ = 323 ± 61.5 ms, mean ± sd; *n* = 9 patches; Fig. 3b and Supplementary Fig. S13). For comparison, we measured the activation/deactivation kinetics of the glutamate sensor SF-iGluSnFR (Supplementary Fig. S13), which showed an even faster fluorescence increase upon application of 1 mM glutamate, as well as fast deactivation kinetics ($\tau$ < 20 ms; Supplementary Fig. S13), in line with previous reports[20]. This confirms that the measured deactivation kinetics of sDarken were not limited by ligand wash-out.

We also characterized the concentration dependence of the sensor kinetics (Fig. 3). At 50 and 100 nM (-$K_d$), the activation kinetics were significantly slowed compared to higher 5-HT concentrations; in contrast, no significant changes were seen in the deactivation kinetics over the whole concentration range (Fig. 3b). In the low concentration regime, the apparent ON kinetics approached the OFF kinetics. The signal changes decreased at 5-HT concentrations <200 nM 5-HT, as expected (Fig. 3b).

The observed kinetic behavior is consistent with a three-state system, in which 5-HT binding is followed by conformational changes, which ultimately cause the fluorescence intensity decrease of sDarken (Scheme Fig. 3 and Supplementary Fig. S14): At high 5-HT concentrations, the observed ON kinetics mainly reflect the conformational changes of the sensor, whereas at lower concentrations, 5-HT binding becomes rate limiting. Fitting of this model showed that the measured kinetics can be described with $k_{12} > 3 \times 10^7 \, \text{M}^{-1} \, \text{s}^{-1}$, $k_{12}/k_{21}$ ~ 1–2 × $10^6 \, \text{M}^{-1}$, $k_{23}$ ~ 20 s$^{-1}$ and $k_{32}$ = 4 s$^{-1}$, respectively (see Supplementary Fig. S14).

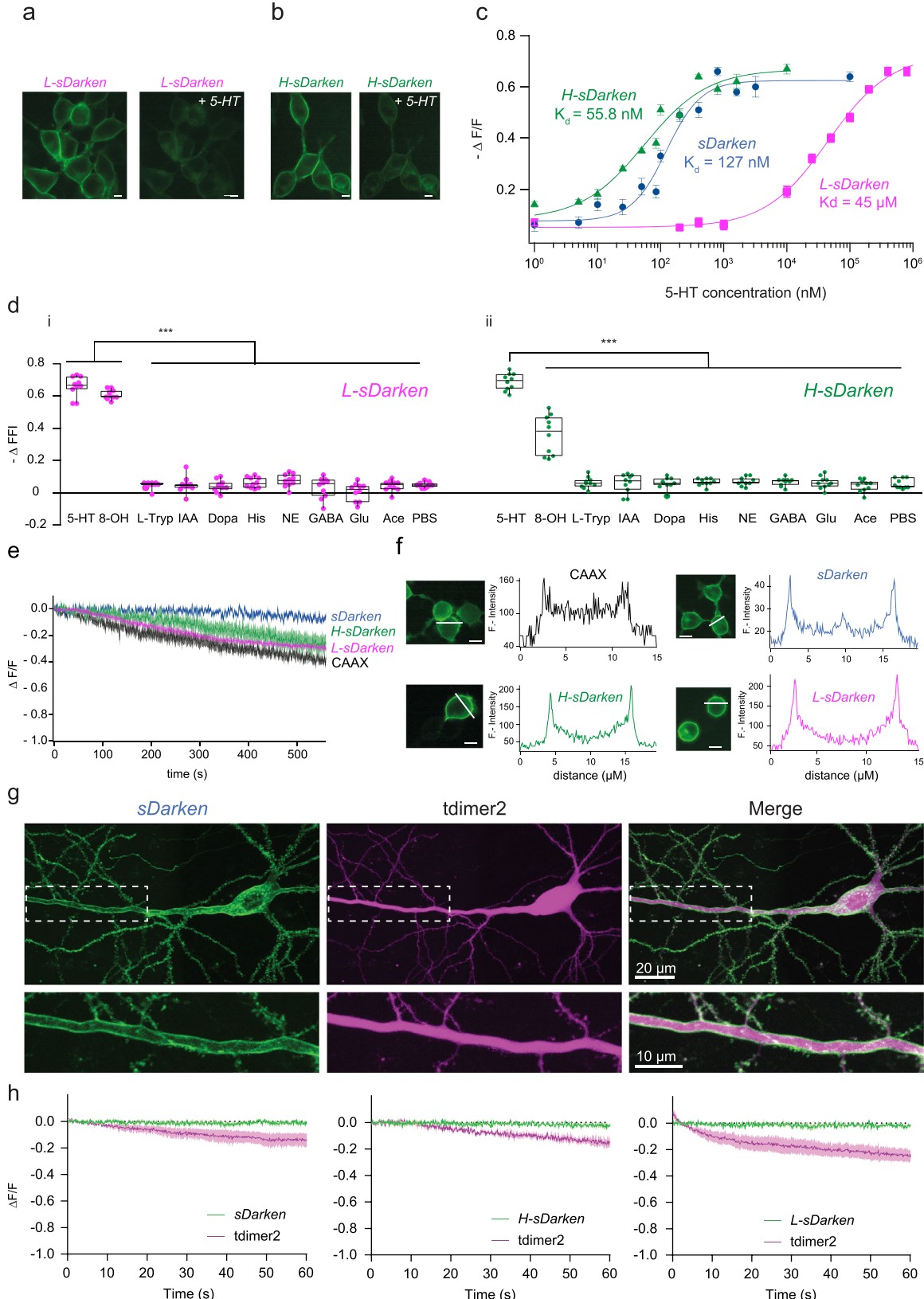

The L-sDarken variant showed slower ON and faster OFF kinetics compared to sDarken ($\tau_{ON} = 95.3 \pm 17.8$ ms and $\tau_{OFF} = 156.0 \pm 27.0$ ms at 100 μM 5-HT; mean ± sd, $n = 4$ patches) (Supplementary Fig. S15a), consistent with its low affinity. H-sDarken showed similar characteristics as the original sDarken ($\tau_{ON} = 57.2 \pm 19.3$ ms and $\tau_{OFF} = 324.8 \pm 48.5$ ms at 1 μM 5-HT mean ± sd, $n = 4$ patches) (Supplementary Fig. S15b).

## 2-Photon imaging in organotypic hippocampal slice cultures

To evaluate the ability of the sensor to report serotonin-binding as observed in HEK cells, we bath-applied serotonin to hippocampal neurons. As expected, all three variants displayed a strong dimming in brightness, measured along dendrites and individual spines, confirming functionality of the sensors in neuronal tissue.

**Fig. 2 | *sDarken* Sensor variants. a** Expression of *L-sDarken* in HEK cells. Fluorescence before and after the application of 160 μM 5-HT. Scale bar 10 μm. **b** Expression of *H-sDarken* in HEK cells. Fluorescence change before and after the application of 800 nM 5-HT. Scale bar 10 μm. **c** Dose response curve of *sDarken*, *L-sDarken* and *H-sDarken* measured in response to different 5-HT concentrations. Group data $n = 20$ from 3 replicates. Error bars represent ±SEM. **d** (i) Fluorescence change of *L-sDarken* to application of serotonin (160 μM), similar substances or neurotransmitters, if not mentioned differently 10 μM were applied ($n = 10$). 8-OHDPAT 300 μM. (ii) Fluorescence change of *H-sDarken* to application of serotonin, similar substances or neurotransmitters if not mentioned differently 10 μM were applied ($n = 10$). Box represents the 25% percentile to the 75% percentile. The

line in the middle of the box represents the median. One-Way ANOVA Multiple Comparison, ***$p < 0.001$. **e** Measurement of photostability. Normalized fluorescence during continuous illumination with blue light (1.8 mW/mm²) for 10 min in HEK cells expressing sensor variants or eGFP-CAAX. **f** Fluorescence intensity profile measured along the white line in one example cell. Scale bar 10 μm. **g** Two-photon image (maximum intensity projection of a *z*-stack) of a CA1 neuron expressing sDarken (green) and the red cytosolic fluorophore tdimer2 (magenta). Insert shows a magnified view of the membrane-localized expression of Darken along the apical dendrite. **h** Time course of *sDarken*, *H-sDarken*, *L-sDarken* and tdimer2 fluorescent signals imaged during 1 min of continuous 2-photon excitation.

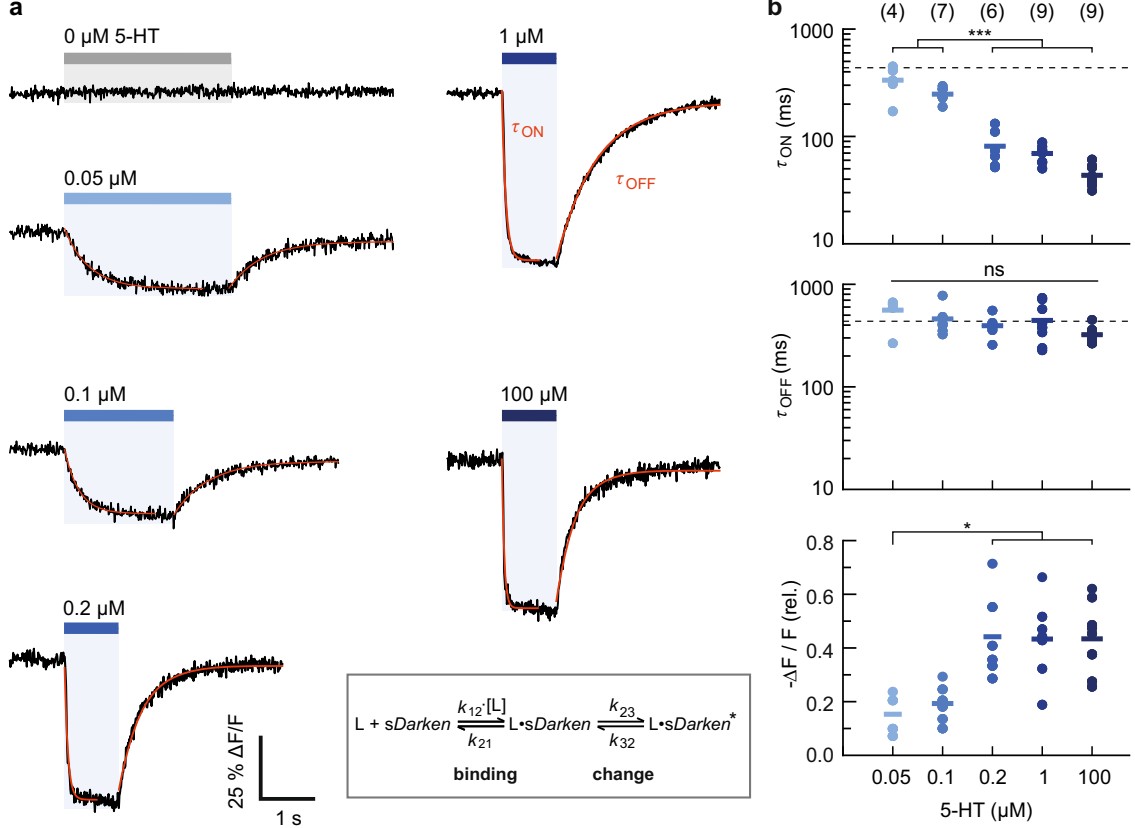

**Fig. 3 | Concentration-dependent kinetics of *sDarken*.** Responses of *sDarken* to increasing 5-HT concentrations, as indicated. **a** Patch fluorometry traces show averages of 6–8 sweeps. ON and OFF time constants were determined by single exponential fits (red lines). **b** The ON kinetics fasten with increasing 5-HT concentrations (top), the OFF kinetics are concentration independent (middle; mean

$\tau_{OFF}$ indicated as dashed line). The signal changes are consistent with $K_d$ ~100 nM (bottom). Number of patches given in parenthesis, means shown as crosses; statistical testing was performed using an ANOVA/Tukey–Kramer procedure, *$p < 0.5$, ***$p < 0.001$. For details on the experiment, analysis and model (inset) see Supplementary Note and Fig. S12.

To emulate the most commonly used method for gene delivery in vivo, we virally transduced the CA1 region of organotypic hippocampal cultures with a recombinant adeno-associated viral vector encoding sDarken. Two to three weeks post rAAV injection, the sensor was strongly expressed and displayed a remarkable membrane localization (Figs. 2g, h and 4a–c). We then measured the fluorescence signal during prolonged 2-Photon raster scanning in control conditions (sham) and after serotonin application (Fig. 4b–e and Supplementary Movies 11 and 12). Again, 2-Photon excitation alone did not induce significant sensor photo bleaching while application of 10 μM serotonin caused a strong decrease in fluorescence detectable both at the somatic and neurophil level (Fig. 4d, e). In additional experiments we expressed the sensors in individual CA1 pyramidal neurons via single-cell electroporation. All three sensor variants showed strong, membrane-localized expression in dendritic spines and axonal terminals (Fig. 4f) and bath application of serotonin

induced a robust decrease in fluorescence in dendrites as well as in spines (Fig. 4f, g).

Taken together, these experiments demonstrate the ability of sDarken to selectively localize in the plasma membrane of hippocampal neurons, to respond to serotonin and its utility for two-photon microscopy.

## Imaging endogenous release in brain slices

Next, we investigated the performance of sDarken in neocortical brain tissue. We expressed sDarken under the neuron-specific synapsin promoter via an AAV injection (pAAV8-hSyn-sDarken) in the prefrontal cortex of mice. We prepared acute brain slices of the infralimbic (Il) region of the prefrontal cortex of mice that expressed sDarken. Exogenous puff applications of serotonin (100 μM) evoked a reliable decrease in the fluorescence signal of sDarken ($\Delta F/F_0 = 0.41 \pm 0.01$), whereas puffs of ACSF did only evoke negligible responses

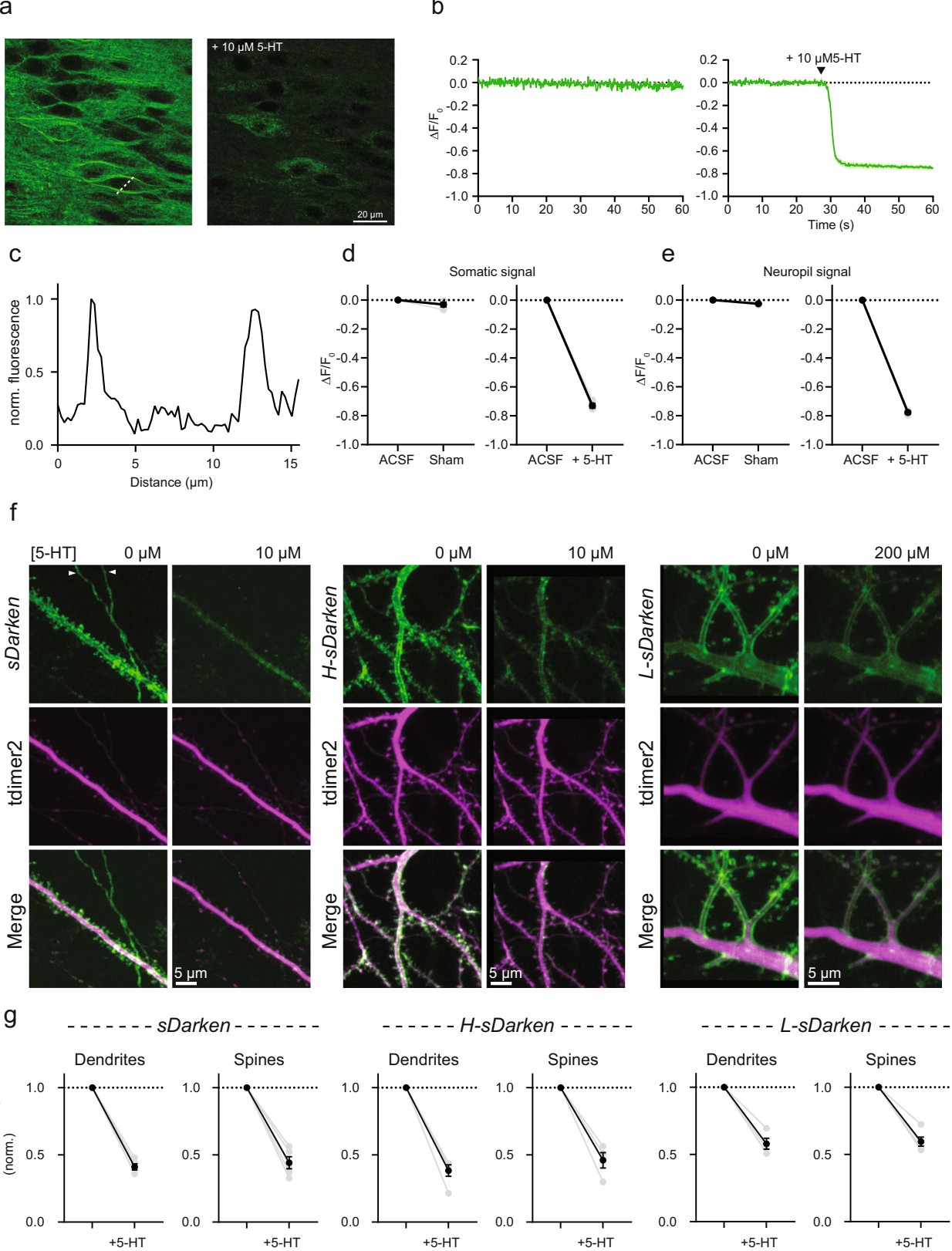

($\Delta F/F_0 = 0.01 \pm 0.008$) (Fig. 5a, b); this also applies to L-sDarken (Supplementary Fig. S16). We also measured the response of sDarken to endogenous release of serotonin. For this purpose, we electrically (40 Hz stimulation) evoked the release of serotonin and imaged sensor dynamics by one photon microscopy (Fig. 5d). Even single pulses evoked a detectable change in fluorescence ($\Delta F/F_0 = -0.042 \pm 0.004$ [1 pulse]) and response amplitudes increased with the number of pulses ($\Delta F/F_0 = -0.05 \pm 0.006$ [2 pulses], $\Delta F/F_0 = -0.05 \pm 0.006$ [5 pulses], $\Delta F/F_0 = -0.08 \pm 0.01$ [10 pulses], $\Delta F/F_0 = -0.076 \pm 0.001$ [20 pulses], $\Delta F/F_0 = -0.13 \pm 0.01$ [40 pulses]) (Fig. 5c, d). In summary, these ex vivo experiments demonstrate that sDarken has sufficient sensitivity to detect endogenous release of serotonin.

**Fig. 4 | *sDarken* and its variants are suitable for 2-Photon applications.**
**a** *sDarken* expression and performance in pyramidal neurons of organotypic hippocampal slice cultures. Two-photon image (single plane) of CA1 neurons expressing *sDarken* in baseline condition (left) and after bath application of 10 μM 5-HT (right). **b** Fluorescence intensity profile of *sDarken* measured along the dotted white line in **a**. Time course of *sDarken* signal imaged in control condition (left) and after bath application of 10 μM 5-HT (right). **c** Normalized fluorescence intensity profile along the dotted line from **a**. **d** Quantification of *sDarken* somatic fluorescence change in control condition (left) and after bath application of 10 μM 5-HT (right) (Sham: $-0.032 \pm 0.013\ \Delta F/F_0$; + 5-HT: $-0.73 \pm 0.014\ \Delta F/F_0$). **e** Quantification of *sDarken* neuropil fluorescence change in control condition (left) and after bath application of 10 μM 5-HT (right) (Sham: $-0.026 \pm 0.003\ \Delta F/F_0$; + 5-HT: $-0.78 \pm 0.007\ \Delta F/F_0$). **f** Expression and performance of *sDarken* (and variants) in subcellular neuronal compartments in response to bath application of serotonin. White arrowheads indicate axonal terminals. **g** Quantification of normalized *sDarken*, *H-sDarken* and *L-sDarken* fluorescence change in dendrites and spines after bath application of 10 μM 5-HT for *sDarken* and *H-sDarken* and 200 μM 5-HT for the high-affinity Darken mutant *L-sDarken* (Darken dendrites: $0.41 \pm 0.02$, $n = 5$ cells; Darken spines: $0.44 \pm 0.04$, $n = 5$ cells; *H-sDarken* dendrites: $0.38 \pm 0.04$, $n = 5$ cells; *H-sDarken* spines: $0.46 \pm 0.06$, $n = 4$ cells; *L-sDarken* dendrites: $0.58 \pm 0.04$, $n = 4$ cells; *H-sDarken* spines: $0.60 \pm 0.03$, $n = 5$ cells). Plots show individual data points (gray filled circles) and average (black filled circles) ±SEM.

## In vivo 2-Photon imaging

To determine if sDarken can detect serotonin dynamics in vivo we employed 2 photon (2P) imaging in the PFC of anesthetized and awake mice, as PFC receives dense projections from the dorsal raphe nucleus (DR), the main nucleus of the serotonergic system[21]. sDarken was injected in the PFC, a cranial window was implanted dorsal to the PFC and a stimulation electrode was placed in the DR (Fig. 5e). sDarken was robustly expressed in superficial and deep cortical layers, in neuropil as well as in somata of the PFC (Fig. 5f). We recorded 5-HT transients during electrical stimulation of the DR in anesthetized mice (Fig. 5f). Electrical stimulation at 50 Hz evoked robust responses in single trials (Fig. 5g and Supplementary Movie 13). Immediately after the onset of electrical stimulation a decrease in fluorescence is apparent, followed by a prolonged increase in fluorescence before returning to baseline intensities (Fig. 5g, h). Serotonin dynamics triggered by electrical stimulation were reliably detected in several individuals (Fig. 5h).

We next wanted to assess the ability of sDarken to detect naturally occurring 5-HT dynamics in the awake animal. Again, sDarken was injected in the PFC and a cranial window was implanted dorsal to the PFC for 2P imaging. For this purpose, we established a reward paradigm (Fig. 6a–e). Mice were trained to run on a linear treadmill (360 cm) to receive oat milk rewards upon licking in three designated reward zones marked by specific textures on the walking surface (Fig. 6b). Mice learned the paradigm, indicated by an increased numbers of licks within the reward zone (Fig. 6d). sDarken was expressed in different cortical layers of the PFC (Fig. 6e–g).

Interestingly, in layer 1 we noticed local serotonin dynamics that occurred independent from locomotion or reward (Fig. 6h). Local events lasted for several seconds to minutes and included smaller transients of 5-HT release. These data suggest that sDarken is sensitive enough to detect natural occurring serotonin dynamics on different timescales. We expected to observe global (full field of view) 5-HT transients during the reward delivery, but this effect was only visible in one out of 3 animals. In this animal fluorescence was decreased (increase in 5-HT) across individual trials and quantification over eleven reward events revealed a significant decrease of fluorescence from baseline (Fig. 6I, j). During data analysis we noticed strong correlation of global 5-HT dynamics with locomotion in all animals ($n = 3$; Fig. 6k, l), a decrease in movement speed was associated with a decrease in fluorescence (increase in forebrain 5-HT) and vice versa a rise in fluorescence (decrease in 5-HT) was evident when the animal increased its movement speed (Fig. 6m, n). Comparison of global sDarken fluorescence during running and rest revealed a robust decrease in fluorescence when animals stopped (Fig. 6l–o). Serotonin dynamics associated with running showed dynamics in the second range, $\tau_{OFF} = 1.24$ s (Fig. 6o). In order to explore serotonin dynamics in more detail we analyzed serotonin dynamics also in cortical layer 2/3 (Fig. 6p and Supplementary Movie 14). In some of the selected ROIs local changes in serotonin (increase in release) proceeded the reward (licks) (Fig. 6p–r), whereas in other no anticipatory change in fluorescence was evident. Our experiments show that sDarken can detect naturally occurring serotonin dynamics on different timescales in the awake animal using 2 P microscopy.

## Discussion

We developed three different genetically encoded 5-HT sensor variants (sDarken), which will enable the imaging of serotonin dynamics in vitro and in vivo. The original sDarken has a high affinity for 5-HT ($K_d$ 127 nM) and fast on and off kinetics ($\tau_{ON} = 43.5 \pm 9.7$ ms an $\tau_{OFF} = 323 \pm 61.5$ ms), making it suitable to detect serotonin dynamics with high temporal and spatial resolution. Complementary to sDarken, two other sensor variants, H-sDarken and L-sDarken, expand the existing toolbox. H-sDarken has a high affinity for serotonin and ($K_d$ 57 nM) and similar kinetics as the original sDarken. L-sDarken has a low affinity for serotonin ($K_d$ $45 \pm 4.6$ μM), slightly slower on kinetics but faster off kinetics. The fast off kinetics will be helpful to avoid buffering of 5-HT and will resolve 5-HT release events at high temporal resolution[3]. Several studies assume that concentration of neurotransmitters within the synaptic cleft can reach millimolar concentrations (>1 mM), whereas extrasynaptic concentrations are expected to be in the high nM to low μM range[22–25]. The affinity of sDarken variants for serotonin is well positioned in this physiological range and our experiments show that sDarken can detect endogenous serotonin release in vitro and in vivo. All sensor variants bind serotonin with high specificity while other neurotransmitters do not elicit fluorescence changes of the sensor. These properties make sDarken to an ideal complement to existing serotonin sensors (Table 1), expanding the toolbox for imaging of serotonin dynamics.

The sDarken family member L-sDarken, with its low affinity and fast temporal resolution, might be suited for the study of point-to-point release sites. In contrast, H-sDarken will be more suitable to detect extracellular volume transmission (Table 1).

In comparison to other recently developed 5-HT sensors (GRAB$_{5-HT}$, PsychLight2 and iSeroSnFR)[10,13,26], which either have fast kinetics or high affinity, sDarken shows high sensitivity combined with relatively fast kinetics (Table 1). Although GRAB$_{5-HT}$ and PsychLight2 use a similar design strategy, i.e., introduction or replacement of IL3, the use of different serotonin receptors as sensing moiety, linkers and insertion sites yielded sensors with distinct properties. For instance, in comparison to PsychLight and sDarken, GRAB sensors do not substitute ICL3 completely, which might lead to residual activation of downstream signaling[3]. This might necessitate the control for undesirable side effects in some research applications. sDarken is the only serotonin sensor, which is bright in the unbound state and diminishes its fluorescence upon binding of serotonin. In comparison to conventional neurotransmitter sensors sDarken has a brightness that is comparable to membrane expression of eGFP. This superior brightness will be favorable for imaging in general. In addition, H-sDarken has a three times higher brightness in comparison to eGFP, well in accordance with properties of superfolder variants[17]. Our experiments also show the potential of sDarken in 2P imaging and its superior membrane expression, brightness and stability. Kinetic properties, in combination with superior brightness and photostability will have advantages in resolving subcellular structures. In the future, specific targeting to synapses could allow for high resolution imaging of serotonin release even at synaptic terminals.

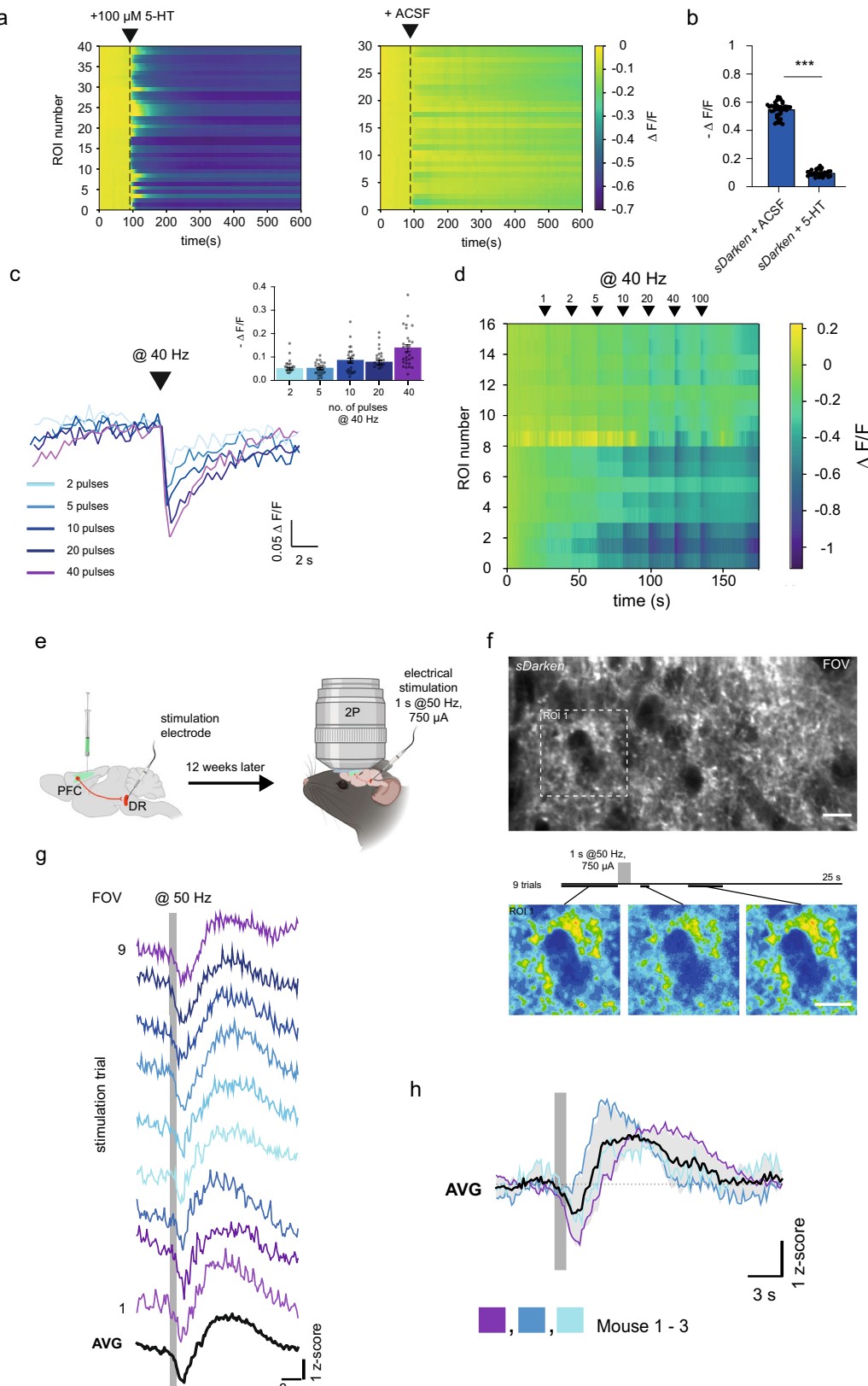

Existing serotonin sensors are either based on 5-HT2 receptors[10,26] or on a mutated periplasmatic binding protein from *E. coli*[13], whereas sDarken is based on the 5-HT1A receptor. 5-HT1A receptors are a major target for anxiolytic, antidepressant, and antipsychotic medications. sDarken could be used in a similar manner as PsychLigh[26] to identify compounds capable of binding to 5-HT1A receptors, that might be useful in finding treatments strategies for psychiatric diseases.

In vivo electrical stimulation in the DR evoked an excessive brief increase of 5-HT levels in the PFC (initial drop in fluorescence) followed by a transient increase in the fluorescence which is most likely arising from autoinhibition of DR[27,28].

**Fig. 5 | Serotonin dynamics in brain slices and in the anesthetized mice. a** Wt mice were injected with pAAV8-Syn-Darken into the prefrontal cortex. 3 weeks after injections brain slices were prepared and imaged with one photon microscopy and a CMOS camera. Left panel puff application of 100 µM 5-HT, $n = 40$ ROIs in 4 slices. Right panel puff application of ACSF, $n = 30$ ROIs in 3 slices. **b** Quantification of fluorescence decrease after puff application from data in **a**, 100 µM 5-HT, $n = 40$ ROIs in 4 slices ACSF, $n = 30$ ROIs in 3 slices, mean and ±SEM are shown, unpaired two-tailed $t$ test, ***$p < 0.0001$. **c** A bipolar stimulation electrode was positioned in the field of view and electrical pulses @40 Hz were delivered (0.2 ms, 750 µA). Single traces for electrical stimulation with different number of pulses inset Quantification of fluorescence responses to the indicated pulse number. $n = 27$ ROIs in 4 slices, mean and ±SEM are shown. **d** Heatmap of analyzed ROIs with increasing pulse numbers. **e** Schematic illustration of the experimental setup: stereotactic injection of AAV-Syn-*sDarken* into the prefrontal cortex (PFC), implantation of a cranial window above the PFC and implantation of a stimulation electrode in the dorsal raphe (DR). Twelve weeks later, anesthetized head-fixed mice were imaged in vivo with 2-Photon imaging. Created with BioRender.com. **f** Upper panel: representative image of in vivo expression of *sDarken* in the PFC (ca. 200 µm deep). Lower panel: timeline of experiment with 9 consecutive electrical stimulations delivered at 50 Hz for 1 s (750 µA, 0.1 ms/pulse) and spaced by 20 s. Underneath, pseudo-colored images depict changes in fluorescence before and after the electrical stimulation. **g** Representative *z*-score transients in the full field of view (FOV) in response to 9 individual electrical stimulations (gray bar) and average (AVG). **h** Average of *z*-score transients in response to 9 electrical stimulations (gray bar) in three mice and grand average of all three mice (AVG).

The same biphasic phenomenon was observed in footshock experiments performed with psychLight2 (another GPCR based serotonin sensor) in the DR and in the prefrontal cortex[26]. Fiber photometry recordings of serotonergic neurons in the DR, when studied as one population, display also biphasic responses to footshock, however, these response properties were thought to originate from anatomically distinct subpopulations[29]. Recent miniscope imaging with single cell resolution, has now impressively shown, that different response types (activation or inhibition) to emotional stimuli, such as reward or footshock, are even present in anatomically defined projections[30]. This heterogeneity could be one explanation to our observed biphasic response in the PFC.

Another explanation might be that the biphasic response is a circuit phenomenon mediated by either autoinhibition within the DR or by top–down projections from the PFC to the DR (activation of a negative feedback mechanism), as biphasic serotonin dynamics in the PFC were only apparent in vivo, whereas upon electrical stimulation in vitro only an increase in serotonin levels was observed. In addition, due to the close proximity to the median raphe (MR) we cannot exclude, that the electrical stimulation activated also MR serotonergic neurons that might contribute to the biphasic response in the PFC[31]. Besides, electrical stimulation might cause an excessive release of serotonin followed by a subsequent depletion of serotonin stores at the synapse, causing a prolonged decrease in serotonergic tone.

Not only artificial (electrical) evoked release of serotonin is detectable with sDarken, also innate changes of serotonin dynamics were visible. Serotonin release is well known to be associated with reward expectation and delivery[32–34], but results obtained in our reward paradigm were ambiguous: Only in one animal (out of two that successfully learned the reward paradigm) a mild, albeit significant increase in serotonin transients (decrease in fluorescence) was visible during reward. Nevertheless, in all animals a strong correlation between 5-HT dynamics and locomotion was evident. This observation fits well with the notion that serotonin in the forebrain is associated with behavioral inhibition, waiting[35,36], and a general suppression of locomotion when 5-HT is released[29,37,38]. It seems, as if serotonin can switch the behavioral state of the animal from active foraging/running to suppression of locomotion[38]. In general, animals showed a mixed strategy to obtain rewards. In some trials mice stopped running during the reward delivery, whereas other animals kept on running during the licking episodes. Furthermore, since we observed a strong modulation of serotonin dynamics by locomotion, it is difficult to isolate serotonin transients that are solely related to rewards. Diverse behavioral strategies of the animals complicate the deciphering of sDarken signals, which might represent differences of the animal´s internal state. Serotonin is a strong modulator of ongoing internal activity and incoming external sensory signals[39–41] and in this light observed serotonin dynamics might also act as gain control for cortical activity.

Interestingly we observed local hot spots, in which 5-HT dynamics were modulated on a slower timescale (minutes), independent from reward or locomotion. Currently, we do not know what these local events encode. Further studies have to reveal, if these local events could be part of a PFC-DR motivational system that regulates effort related behaviors in challenging situations[42,43].

Our experiments demonstrate the ability of sDarken to image naturally occurring serotonin dynamics on a physiological and behavioral relevant timescale (seconds) by 2P imaging in vivo. We deliberately decided to use 2P imaging, as, in contrast to fiber photometry, 2P imaging has high requirements for the signal-to-noise ratio and a high sensitivity of fluorescent sensors due to possible motion artifacts, hemodynamics and imaging depth[3]. Fiber photometry on the other hand is based on time-correlated single-photon counts and is an ultrasensitive tool to detect emitted light[2], however, it lacks cellular resolution. With sDarken, serotonin release was even detectible during single trials in 2P imaging and the fast off-rates enabled the precise measurement of serotonin transients in vivo. Our experiments revealed a strong spatial variation in serotonin volume transmission, manifested in small localized hotpots, in which serotonin dynamics were evident anticipatory before the delivery of the reward. Similar observations have been made for dopamine release in the cortex[8].

Future experiments will reveal if sDarken also allows for the visualization of serotonin release at dendrites, single spines or axons. Measurements in organotypic cultures and 2P measurements with cellular resolution in vivo suggest that also subcellular resolution, i.e., boutons and synapses, will be possible with sDarken.

Obviously, the expression of sensor proteins for neurotransmitters can distort their diffusion/reuptake dynamics and the signals they elicit by buffering the analyte (see e.g., ref. 44). These effects can be circumvented by choosing a variant with an ideal affinity range (sDarken, L-sDarken, or H-sDarken) and by keeping the expression levels as low as possible. The strong fluorescence and large signal changes of the sDarken family is particularly helpful in that regard. At the same time, low expression levels will reduce possible assembly with native 5-HT1A or other receptors into homo- or heterodimers, respectively, or interactions with other scaffolding proteins. Other biosensors, that utilize the same design principle as sDarken, did not show any significant effects on endogenous 5-HT receptor signaling[10].

The sDarken family offers three variants with high specificity, a broad dynamic range, high affinity (nM–mM) and improved temporal resolution in combination with cellular and molecular specificity. Due to their high specificity and photostability, they are well suited to investigate serotonin dynamics in the behaving animal.

We think that the sDarken family will enlarge the available toolbox for genetically encoded serotonin sensors. sDarken will allow the investigation of 5-HT transients and serotonin dynamics in behavior and in different brain states. Designed sensor variants will greatly facilitate the general understanding of 5-HT neurotransmission. We expect further improvements in sensor properties and the development of sensors with different fluorophores in the future.

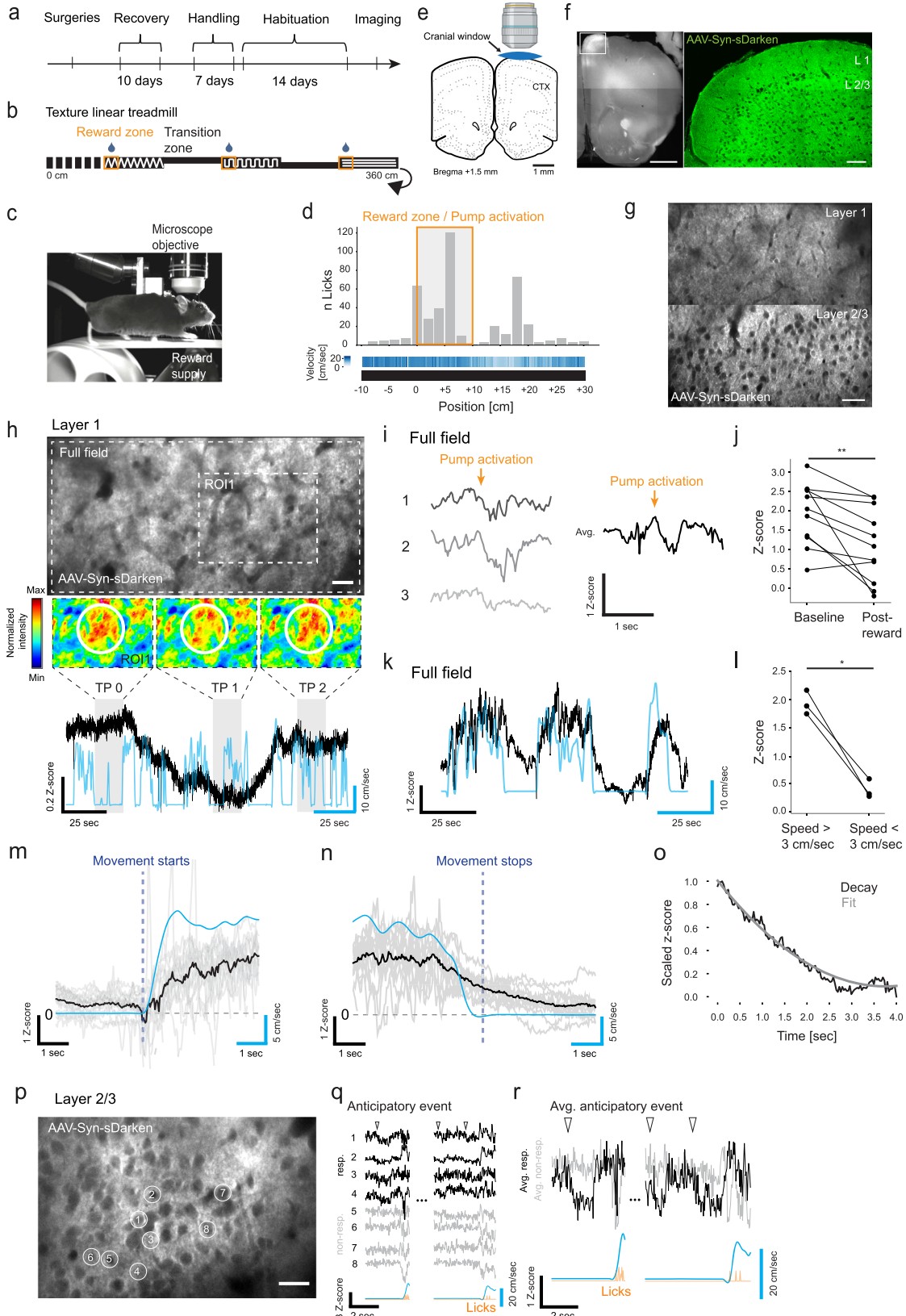

## Methods

Our research complies with all relevant ethical regulations and have been approved by local authorities: Freie Hansestadt Bremen, Senatorin für Gesundheit, Frauen und Verbraucherschutz and LANUV Germany.

### Cloning

pN1-GCaMP6m-XC was a gift from Xiaodong Liu (Addgene plasmid # 111543; http://n2t.net/addgene:111543; RRID:Addgene_111543) and sfGFP-N1 was a gift from Michael Davidson & Geoffrey Waldo (Addgene plasmid # 54737; http://n2t.net/addgene:54737; RRID:Addgene_54737).

**Fig. 6 | Two-photon in vivo imaging. a** Time schedule. **b** Depiction of linear treadmill. Reward positions are indicated by orange box. **c** Representational image of head-fixed animal running on linear treadmill. **d** Number of licks over spatial bins (gray bars) of mice ($n = 2$). Heatmap indicates averaged velocity over spatial bins. **e** Schematic representation of cranial window above PFC. Created with BioRender.com. **f** Histological images of cortical AAV-Syn-sDarken expression. Left image: Overview of ipsilateral hemisphere (Scale bar = 1 mm). Right image: Blow up of layer 1 and layer 2/3 (Scale bar = 100 μm). **g** Two-photon imaging field-of-view of AAV-Syn-sDarken expression in PFC layer 1 (top) and layer 2/3 (bottom). (Scale bar = 50 μm). **h** Large image shows average intensity projection of *sDarken* expression in dendritic layer in PFC (Scale bar = 100 μm). Small images show 16-colors intensity projection of ROI1 at different time intervals (TP0–TP2). Black trace shows fluorescence depicted as *z*-score, blue trace shows velocity of mouse. **i** Three exemplary traces of full field fluorescence changes in response to reward. Right trace average of 11 reward events in one animal. **j** Quantification of average *z*-score

at (median = 2.76, $n = 11$ reward events in one animal) and post-reward (median = 2.10, $n = 11$). $P < 0.01$, Wilcoxon matched-pairs signed ranks test. **k** Black trace full-field sDarken fluorescence, blue trace velocity of mouse. **l** Quantification of average fluorescence during periods of running (speed >3 cm/s) and periods of rest (speed <3 cm/s) (running: mean = 1.94; rest: mean = 0.42; $n = 3$ mice). $P < 0.05$, Paired $t$ test. **m** Gray traces single fluorescence traces ($n = 16$) related to movement starts ($n = 3$). Black trace shows averaged trace. **n** Gray traces show single fluorescence traces ($n = 18$) related to movement stops ($n = 3$). Black trace shows averaged trace. **o** Black trace shows min-max-scaled averaged fluorescence decay after movement stops. Gray line indicates 2nd-degree polynomial fit of fluorescence decay (tau$_{off}$ = 1.24 s). **p** Average intensity projection of cellular layer 2/3 in PFC (Scale bar 25 μm). **q** *Z*-scored fluorescence traces of responding (black) and non-responding ROIs (gray) preceding reward-seeking behavior (orange) and initiation of movement (light blue). **r** Average z-score of responding ROI 1–4 (black) and non-responding ROI 5–8 (gray).

### Table 1 | Comparison of genetically encoded serotonin sensors

| Sensor | ($\Delta F/F_0$) % change | | | | | | $K_d$ | $\tau_{ON}$ (ms) | $\tau_{OFF}$ = (ms) |
|---|---|---|---|---|---|---|---|---|---|
| | 10 nM | 100 nM | 1.6 μM | 3.2 μM | 100 μM | 1 mM | | | |
| L-sDarken | – | – | – | –20 | –48 | –73 | 45 μM | 95 | 156 |
| sDarken | –14 | –33 | –58 | –60 | –40 | Sat. | 127 nM | 43 | 323 |
| H-sDarken | –8 | –51 | –62 | –67 | Sat. | Sat. | 57 nM | 57 | 324 |
| GRAB5-HT1[10] | 70 | 240 | 280 | Sat | Sat | Sat | 22 nM | 200 | 3100 |
| iSero[13] | 15 | 20 | 30 | 75 | 750 | Sat | 390 μM | 0.5–10 (fast) 5–18 (slow) | 4 (fast) 150 (slow) |
| PsychLight2[26] | 25 | 55 | 80 | Sat | Sat | Sat | 26.3 nM | | 997 (fast) 3998 (slow) |

DNA fragments were amplified by PCR using the CloneAmp HiFi Polymerase (Clonetech) with Primers (Thermo Fischer Scientific) containing Overhangs of ~20 bp complementary to the ends of each other PCR products necessary for plasmid generation.

Plasmids were then generated with the use of the InFusion cloning (Clonetech) method and purified using a Nucleo Bond Xtra Midi Kit (Macherey Nagel). To verify the correct assembly and DNA sequence, Sanger Sequencing was performed by LGC genomics with standard primers provided by them (CMV-F, EGFP-C1-R). For experiments of random mutations in the Linker sequence, Primers were used, containing overhangs for the InFusion reaction and NNK triplets (N = any base, K = G or T) in specific sites selected for the mutations. Products were purified using the GeneJet Plasmid Mini Prep Kit (Thermo scientific).

### Cell culture
Human embryonic kidney 293 (HEK293) cells (tsA201 cells, ATTC) and stably expressing GIRK1/2 subunits HEK293 (gift from Dr. Andy Tinker (UCL London, UK) cells were cultured in DMEM (TSA Cells: DMEM, FBS 10%, Penicillin/Streptomycin 10,000 U/ml; GIRK1/2 Cells: DMEM, FBS 10%, Penicillin/Streptomycin 10,000 U/ml, G418 400 μg/ml) in 25-cm² flasks at a temperature of 37 °C and $CO_2$ concentration of 5%. Cells were transfected with 3 μg/ml PEI and 0.5 μg plasmid DNA incubated for 18 –24 h before recordings.

For Imaging experiments, cells were splitted into 60 mm² dishes and transfected, after reaching 60-70% confluency, 18–24 h before recordings using PEI (3 μg DNA with 10 μl Pei 1 μg/μl in 150 μl DMEM, incubation for 15 min).

### Characterization in HEK cells
**Imaging in HEK cells.** Prior to imaging cells were washed with PBS.Imaging was performed at an upright LNscope from Luigs&Neumann using a 40x water immersion objective (LUMPLFLN40xW, Olympus).

Videos were recorded with a CMOS camera (Hamamatsu) at a framerate of 1 Hz for 2 min.

Neurotransmitter and drugs were directly applied to the bath with a gravity driven perfusion system.

Videos were analyzed in ImageJ {Schneider:2012dw. ROIs were selected around the cell membrane. Mean gray (fluorescence) values for each timepoint were measured. Obtained values were corrected by subtracting background fluorescence. Data were normalized to the first frame by dividing all mean values by the according basal fluorescence ($F_{base}$) of the cell. To obtain $\Delta F/F$ values, we subtracted baseline fluorescence prior ($F_0$) to ligand application with fluorescence after application of the ligand ($F$).

$$\Delta F/F = \frac{F - F_0}{F_0}$$

For titration curves, data were fitted in Igor Pro(WaveMetrics) with a Hill function.

For a better comparison to existing positive going sensors, we report the typical decrease in fluorescence of *sDarken* as positive values for $K_d$ plots and agonist specificity.

### Uncaging of serotonin
HEK293T cells were seeded in 35 mm culture dishes and subsequently transfected with either one of the three 5-HT sensor variants (i.e., *L-sDarken*, *sDarken*, *H-sDarken*). Imaging was performed 24 h post-transfection. For this, the culture medium was replaced by PBS and 100 μM of NPEC-caged-serotonin (Cat. No. 3991, Tocris) was added to the dish. HEK293T cells expressing the 5-HT variants were imaged on a laser scanning confocal Zeiss LSM 880. We used the 488 nm laser for imaging the fluorescence of the 5-HT sensor variants. Imaging was performed using ×60 oil immersion objective (LD LCI Plan-Apochromat ×63/1.2) at an imaging speed of 0.93 Hz. For uncaging of the NPEC-caged-serotonin we used the 405 nm laser at 90% power. The bleaching area (i.e., uncaging area) was selected manually via the ZEN software of the LSM. The bleaching period was set to 90 ms and the interval between two bleaching periods was set to 50 s.

### Electrophysiology
For GIRK channel recordings native 5-HT$_{1A}$ receptors and 5-HT sensor variants were expressed in HEK293 cells stably expressing GIRK1/

2 subunits. Cells were cultured and recorded in dark room conditions after transfection. To avoid overactivation and internalization of transfected native 5-HT$_{1A}$ receptors 5-HT containing DMEM was replaced by dialyzed DMEM before transfection. Transfected cells were transferred to poly-l-lysin (0.001%) coated 15 mm coverslips 2–3 h prior to measurements. Only separated cells were used for patch clamp recordings. GIRK-mediated K+ currents were measured and analyzed as described previously[45]. The external solution was as follows: 20 mM NaCl, 120 mM KCl, 2 mM CaCl$_2$, 1 mM MgCl$_2$, 10 mM HEPES-KOH, pH 7.3 (KOH). Patch pipettes (2–5 megaohms) were filled with internal solution: 100 mM potassium aspartate, 40 mM KCl, 5 mM MgATP, 10 mM HEPES-KOH, 5 mM NaCl, 2 mM EGTA, 2 mM MgCl$_2$, 0.01 mM GTP, pH 7.3 (KOH).

Cells were visualized using a trans-illuminated red light (590 nm) or blue light filter (450 nm) during experimental manipulations. Whole-cell patch clamp recordings of HEK293 cells were performed with an EPC10 USB amplifier (HEKA). Currents were digitized at 10 kHz and filtered with the internal 10-kHz three-pole Bessel filter (filter 1) in series with a 2.9-kHz 4-pole Bessel filter (filter 2) of the EPC10 USB amplifier. Series resistances were partially compensated between 70 and 90%. Leak and capacitive currents were subtracted by using hyperpolarizing pulses from −60 to −70 mV with the p/4 method. Also, 50 Hz noise was canceled by a Hum Bug filter. Current responses were quantified with the software Igor Pro (Wavemetrics).

## GPCR assays

To analyze the potential activation of the Gα intracellular signaling cascades (i.e., Gq and Gs) by the binding of 5-HT to the 5-HT sensor variants, we used two similar approaches. First, HEK293T cells were seeded in 35 mm culture dishes and subsequently transfected with either one of the three 5-HT sensor variants and jRCaMP or CNG-jRCaMP overnight. For analyzing potential Gq-pathway activation we recorded jRCaMP fluorescence during 5-HT stimulation, since activation of the Gq-pathway results, among others, in an increase in intracellular Ca$^{2+}$ levels. As a positive control we subsequently stimulated the cells with ATP, which led to a rise in intracellular Ca2+ levels. For analyzing potential Gs-pathway activation we recorded jRCaMP fluorescence within HEK293tsa cells, which were co-transfected with a cyclic nucleotide–gated ion channel (CNG) (Kindly provided by Dr. Reinhard Seifert from the Core Facility for Genetic Engineering of the Research Center Caesar), during 5-HT stimulation. CNGs drive a Ca$^{2+}$ influx upon binding of cAMP. Therefore, Gs-Pathway activation would increase cAMP production through the activation of the adenylate cyclase and would subsequently lead to increase intracellular Ca$^{2+}$, measured by an increase in jRCaMP fluorescence. As a positive control we stimulated the cells with Forskolin, which activates the adenylate cyclase and therefore increases intracellular cAMP levels.

## Influence of different pH values on the fluorescence of *sDarken*

First, HEK293T cells were seeded in 35 mm culture dishes and subsequently transfected with sDarken. Imaging was performed at an upright LNscope from Luigs&Neumann using a 40x water immersion objective (LUMPLFLN40xW, Olympus). The next day cells were imaged (see above for detailed description) in PBS, set to different pH values (6.0–8.0; 0.2 steps), either with or without 10 μM of 5-HT. Furthermore, HEK293T cells expressing *sDarken* were tested for their fluorescence change due to 5-HT wash in, in different pH environments. After 30 s of baseline measurement in PBS set to different pH values (6.2-7.8; 0.4 steps) PBS + 100 μM 5-HT, also set to the same pH value, was washed in. $\Delta F/F$ was calculated as described in (imaging in HEK cells). For the analysis of the pre-post difference, we subtracted a 5 s average (second 20-25) before the start if the 5-HT wash in from a 5 s average (second 70–75) after the start of the 5-HT wash in.

Videos were recorded with a CMOS camera (Hamamatsu) at a framerate of 1 Hz. To evaluate if changes in pH alone can elicit fluorescence changes of the sensor, we washed in modified Ringer buffer (140 mM NaCl, 3.5 mM KCl, 0.5 mM NaH$_2$PO$_4$, 0.5 mM MgSO$_4$, 1.5 mM CaCl, 10 mM HEPES, 2 mM NaHCO$_3$, 5 mM Glucose) set to 4 different pH values (6.8, 7.0, 7.2, 7.4) in a random order, as a consecutive trial. Data was normalized to the first frame for every trial. Evaluation of the influence of pH on the dose-response curves of the sensor we measured changes in fluorescence ($\Delta F/F$) for 7 different concentrations (10 nM, 50 nM, 100 nM, 200 nM, 400 nM, 1 μM, 10 μM) and 4 different pH values (6.8, 7.0, 7.2, 7.4). Titration curves data were fitted in GraphPadPrism (9.3.1) with a specific binding with Hill slope.

## Kinetic characterization by fast patch-clamp fluorometry

5-HT sensor (*sDarken*) kinetics were analyzed by fast ligand applications to outside-out membrane patches in combination with fluorescence imaging, i.e., by fast patch-clamp fluorometry.

## Sensor expression and patch-clamp procedure

HEK 293T cells (DSMZ (German Collection of Microorganism and Cell Cultures), #ACC 635) were grown in DMEM with 8% FBS at 37 °C and 5 % CO$_2$ on plastic (PET-G) coverslips. Transfections were conducted after 24-48 h using polyethylenimine 25,000 with -0.3 μg DNA per ml medium. Membrane patches were pulled after 48 h expression using standard patch-clamp procedures. In brief, patch pipettes (3–4 MΩ resistance) were pulled from borosilicate glass (G150TF-4, Warner Instruments) and filled with internal solution (122 mM CsCl, 2 mM NaCl, 2 mM MgCl$_2$, 10 mM EGTA, 10 mM HEPES, pH 7.2). Coverslips with adherent cells were placed in external solution (138 mM NaCl, 1.5 mM KCl, 1.2 mM MgCl$_2$, 2.5 mM CaCl$_2$, 10 mM HEPES, pH 7.3). Using a micro-manipulator (Patchstar, Scientifica) and an Axopatch 200B patch-clamp amplifier (in combination with a Digidata 1550 A/D converter and pClamp 10.7 software; all Molecular Devices) cell-attached and then whole-cell configurations were established to excise outside-out patches. Experiments were performed at 22–25 °C with outside-out patches voltage-clamped to −70 mV. The resistance was typically ≥1 GΩ and was monitored throughout the experiment. pClamp and the A/D converter were used for generating the voltage pulses for ligand application and triggering the acquisition of single camera frames. Chemicals, including 5-HT hydrochloride and glutamate, were purchased from Sigma. Control experiments were performed with pAAV SFiGluSnFR.A184V (Addgene #106199)[46].

## Fast ligand application and imaging

Fast ligand application and removal was achieved by positioning the outside-out patches in front of a piezo-driven double-barreled Θ-glass pipette. The pipette was pulled from borosilicate glass (OD 2.0 mm, ID 1.40 mm, septum 0.2 mm, Warner Instruments), broken to yield a -150 μm diameter tip, and mounted to a piezo actuator (P842.20, Physik Instrumente). Lateral displacements were triggered with short voltage steps (3 V ramp in 0.7 ms), amplified by a power supply (E505.10, Physik Instrumente) and filtered at 1 kHz. Solutions were delivered using a syringe pump (0.2-0.4 ml/min per channel) with parallel bath perfusion at ~3-5 ml/min. Exchange currents between extracellular solution 0.5× and extracellular solution 1× confirmed sub-millisecond solution exchange (Fig. S10). Typically, 5-HT was applied for 1 s in 6–8 subsequent sweeps. For lower concentrations, the application time was increased up to 4 s.

Epifluorescence imaging was performed on an inverse microscope (DMi8, Leica), with a ×40 objective (HCX PL FL L 40x/0.60 CORR XT). Green fluorescence was excited using a 470 nm LED (Thorlabs) and a 470/40 nm excitation filter in combination with a 495 nm dichroic mirror and a 525/50 nm emission filter (all Chroma). The light intensity in the focal plane was -10.4 mW/mm$^2$. Images were acquired with an EMCCD camera (Evolve 512 delta, Photometrics) using Micro-Manager2.0. The acquisition of individual frames was controlled by TTL pulses (presequence/strobed mode) triggered via pClamp 10.7

(Molecular Devices). Imaging of the 5-HT sensor (Figs. 1e, 3, S11, and S13) was performed with an effective frame rate of 91 Hz using an exposure time of 10 ms, a gain setting of 50, and cropping to a region of 9 × 16 pixels. SFiGluSnFR(A184V) (Fig. S11) responses were measured with an exposure time of 5 ms with an effective frame rate of 194 Hz.

## Data analysis
Experiments were repeated several times after independent transfections. ImageJ 1.53c ([47]Rasband W./NIH) was used to extract the fluorescence intensity of a defined region (9–16 pixel oval) of the membrane patch and a neighboring region for background subtraction. The data were transferred to Clampfit 7 (Molecular Devices), baseline corrected (linear adjustment) and averaged for further analysis. Fluorescence changes, $\Delta F/F$ (%), and time constants from single exponential fits, $\tau_{ON}$ and $\tau_{OFF}$, (Fig. 3b) were determined from averaged traces (typically 6–8 sweeps) by least-square fitting using ProFit 7.0 (Quantumsoft). Single exponential fits provided a reasonable description of all traces, although a minor slow phase was visible in some cases. These slow phases, typically present in the ON and OFF kinetics, varied between patches and are likely to reflect inhomogeneous perfusion in the patch pipette. Statistical analysis was performed with Excel (Microsoft) and Statistica 13.3 (StatSoft). All data sets showed normal distribution (Shapiro–Wilk tests, $p < 0.05$). One-way ANOVA ($p < 0.05$) followed by a Tukey–Kramer post hoc testing procedure was used to compare up to five conditions (Fig. 3b). Pairwise comparison was performed using Welch's $t$ test (Fig. S15). Calculations on the kinetic model (Fig. S14) were performed with ProFit 7.0 (Quantumsoft).

## Slice culture preparation and transgene delivery
Organotypic hippocampal slices were prepared from Wistar rats at post-natal days 5–7 as previously described. Briefly, dissected hippocampi were cut into 400 µm slices with a tissue chopper and placed on a porous membrane (Millicell CM, Millipore). Cultures were incubated at 37 °C, 5% $CO_2$ in a medium containing 80% MEM (Sigma M7278), 20% heat-inactivated horse serum (Sigma H1138) supplemented with 1 mM L-glutamine, 0.00125% ascorbic acid, 0.01 mg/ml insulin, 1.44 mM $CaCl_2$, 2 mM $MgSO_4$ and 13 mM D-glucose. No antibiotics were added to the culture medium.

For plasmid delivery in organotypic slices, individual CA1 pyramidal cells were transfected by single-cell electroporation between DIV 15 and 20 as previously described[48]. The plasmids pAAV-syn-sDarken, pAAV-syn-H-sDarken and pAAV-syn-L-sDarken were all diluted to a concentration of 50 ng/µl while the pCI-syn-tdimer2 plasmid was diluted to a concentration of 10 ng/µl. All plasmids were diluted in K-gluconate-based solution consisting of (in mM): 135 K-gluconate, 10 HEPES, 0.2 EGTA, 4 $Na_2$-ATP, 0.4 Na-GTP, 4 $MgCl_2$, 3 ascorbate, 10 $Na_2$-phosphocreatine, pH 7.2, 295 mOsm/kg. An Axoporator 800 A (Molecular Devices) was used to deliver 25 hyperpolarizing pulses (−12 V, 0.5 ms) at 50 Hz. During electroporation slices were maintained in pre-warmed (37 °C) HEPES-buffered solution in (mM): 145 NaCl, 10 HEPES, 25 D-glucose, 2.5 KCl, 1 $MgCl_2$ and 2 $CaCl_2$ (pH 7.4, sterile filtered).

For viral vector-based transduction of organotypic hippocampal slice cultures, AAV particles encoding rAAV8-syn-Darken were pressure injected (20 PSI/2-2.5 bar, 50 ms duration) using a Picospritzer III (Parker) under visual guidance (oblique illumination) into CA1 stratum pyramidale between DIV 3-5. Slice cultures were then maintained in the incubator for another 2–3 weeks to allow construct expression.

## Two-photon microscopy in organotypic hippocampal slice cultures
All the organotypic hippocampal slice culture experiments were performed at room temperature (21–23 °C) between DIV 21 and 23. The slices were submerged in artificial cerebrospinal fluid (ACSF) consisted of (in mM): 135 NaCl, 2.5 KCl, 4 $CaCl_2$, 4 $MgCl_2$, 10 Na-HEPES, 12.5 D-glucose, 1.25 $NaH_2PO_4$ (pH 7.4) under a custom-built two-photon imaging setup based on an Olympus BX51WI microscope controlled by modified version of ScanImage 3.8. Two tunable, pulsed Ti:Sapphire laser (MaiTai DeepSee, Spectra Physics) controlled by an electro-optic modulator (350-80, Conoptics) tuned to 930 and 1040 nm was used to excite Darken (and variants) and tdimer2, respectively. Emitted photons were detected through the objective (LUMPLFLN 60XW, ×60, 1.0 NA, Olympus) and through the oil immersion condenser (numerical aperture 1.4, Olympus) by photomultiplier tubes (H7422P-40SEL, Hamamatsu). 560 DXCR dichroic mirrors and 525/50 and 607/70 emission filters (Chroma Technology) were used to separate green and red fluorescence. Excitation light was blocked by short-pass filters (ET700SP-2P, Chroma).

To visualize sDarken fluorescence change, serotonin hydrochloride (Tocris, UK) was bath applied at a concentration of 10 µM for sDarken and H-sDarken and 200 µM for L-sDarken. For post hoc analysis of sDarken fluorescence change, we draw ROI along stretches of expressing dendrites and spines and measured fluorescence values of maximum intensity projections images, acquired before and after serotonin application. Post serotonin fluorescence values were divided by pre serotonin (baseline) fluorescence to obtain normalized fluorescence change values.

For the relative change in sDarken (and variants) fluorescence ($\Delta F/F_0$) measurements during live imaging experiments, we compared the average relative fluorescence values of the first 40 frames (1 frames = 128 ms) to the average relative fluorescence values of the last 40 frames (for both sham and +5-HT condition). In total 470 frames (128 × 128 pixels) were acquired at 7.8 Hz. We selected ROIs at the cell somatic membrane and from the fluorescence signal of the neuropil in stratum radiatum. Analysis was performed in ImageJ[47].

## Virus production
Adeno associated virus of the serotype 8 (AAV8-hSyn-sDarken) were produced in house. HEK-T-cells were transfected with the help of the AAV Helper-Free System (Agilent) and after three days harvested and concentrated the Virus with the AAVanced™ Concentration Reagent (System Biosciences).

## Imaging in acute brain slices
For acute brain slice experiments C57BL/6 J mice were deeply anesthetized with i.p injection of Ketamine (0,12 mg/g) and Xylazin (0,016 mg/kg) and placed in a stereotaxic frame. AAV particles encoding rAAV8-syn-sDarken or rAAV8-L-sDarken) were bilateral injected into the prefrontal cortex of mice cortex via pressure injection using customized glass pipettes (AP: 1.7 mm; ML: 0.3 mm; DV: −2.2 mm, −2.0 mm–1.8 mm). After surgery mice received subcutaneous Carpofen (2 mg/g) as analgesia. Three weeks after virus injection mice were decapitated and brains were removed. 300 µm coronal slices of the PFC were cut with a vibratome (7000 SMZ, Campden Instruments) in ice cold oxygenated dissection solution containing 87 mM NaCl, 2.5 mM KCl, 75 mM Sucrose, 1.25 mM $NaH_2PO_4$, 25 mM $NaHCO_3$, 7 mM $MgCl_2$, 0.5 $CaCl_2$, 20 mM D( + )-Glucose, bubbled with 5% $CO_2$, 95 % $H_2O$. Slices were allowed to recover for 30 min in 36 °C oxygenated dissection solution and incubated for 1 hour in 36 °C oxygenated ACSF containing 124 mM NaCl, 2.5 mM KCl, 1.25 mM $NaH2PO_4$, 2 mM $MgSO_4$, 26 mM $NaHCO_3$, 2.5 mM $CaCl_2$, 10 mM D( + )-Glucose, 4 mM L-ascorbic acid, bubbled with 5% $CO_2$, 95% $H_2O$. Afterwards slices were stored in oxygenated ACSF at RT till start of experiment.

For imaging slices were transferred to a recording chamber that was continuously perfused with room tempered oxygenated ACSF. Measurements were performed at an upright Olympus microscope using a ×40 water immersion objective (LUMPLFLN40xW). Video were recorded with a CMOS camera (Hamatsu) at a framerate of 5 Hz and an exposure time of 0, 1 s.

To measure the response of *sDarken* to exogenous serotonin 10 μM 5-HT was applied using a customized borosilicate glass pipette (Harvard Apparatus, ~2–4 μm diameter). To image the response of *sDarken* to endogenous serotonin release, 5-HT terminals within the prefrontal cortex were electrically stimulated using a bipolar stimulation electrode (Lohmann Research Equipment, Dortmund, Germany). Stimulation pulses of 750 μA @40 Hz with increasing number of pulses (1, 2, 5, 10, 20, 40, 80, 100) were applied using a four-channel stimulus generator (STG 4004, Multichannel Systems, Reutlingen, Germany) which was controlled by the Synchrobrain Software (Lohmann Research Equipment, Dortmund, Germany).

### In vivo two-photon imaging

**Virus-injection, cranial window and electrode implantation.** For virus-injection and implantation of cranial windows and electrodes mice were anesthetized and fixed on a stereotactic frame. Body temperature was controlled by a rodent heating pad (WPI) and eyes protected by ophthalmic ointment (Bepanthen®, Bayer). After application of lidocaine, the dermis on top of the skull was removed in a circular shape. The bone was made porous using phosphoric acid (OptiBond™ FL bottle kit, Kerr) before covering the skull in a light-cured two-component base (OptiBond™ FL bottle kit, Kerr). The PFC was exposed with a 4 mm circular craniotomy and the Dura mater was carefully removed. For the injection of AAV-Syn-sDarken the virus was loaded into a microliter syringe (Hamilton) equipped with a NanoFil needle (NF35BV, World Precision Instruments). The syringe was lowered into the brain at four locations to inject virus into the PrL (AP + 2.7 mm; ML ± 0.25 mm; DV −0.3 mm) as well as ACC (AP + 1.6 mm; ML ± 0.25 mm; DV −0.4 mm). A volume of 250 nl virus (100 nl/min) was injected at each location using a four-channel micro controller (SYS-Micro4, World Precision Instruments) and allowed to diffuse for 5 min before syringe was slowly removed. After finalizing the virus-injection, the craniotomy was covered with a 4 mm cover slip (thickness 1) and sealed with light-curable composite (GRADIA® DIRECT Flo, GC Europe). Following the window implantation, the tip of a bipolar twisted platinum electrode (PlasticOne) was stereotactically placed into the dorsal raphe nucleus (AP −4.7 mm; ML ± 0.0 mm; DV −2.2 mm) and fixed with the light-curable composite. Finally, a metal bar (Luigs and Neumann) for head-fixation was added and the whole skull covered in light-curable composite. After surgery mice were placed in a heating chamber and allowed to wake up. Buprenorphine was administered for 3 days post-surgery.

### In vivo imaging in head-fixed mice

10 days after surgeries, mice were handled and habituated to the reward (oat milk) for 5 days. Following, mice were habituated to the linear treadmill (Luigs and Neumann), head-fixation and learned to associated three reward zones (10 cm) per lap on the linear treadmill (360 cm) with the reception of the reward for two weeks. Subsequently, timelapses showing changes in the fluorescence of sDarken were acquired while mice were moving voluntary on the treadmill and receiving rewards in the learned reward zones. For acquisition a custom-made Thorlaps two-photon microscope connected with a titanium sapphire 80 MHz Chameleon Ultra II two-photon laser (Coherent, Inc.) and equipped with an 8 kHz galvo-resonant scanner (LSK.GR08/M, Thorlabs), a GaAsP PMT (Thorlabs) and a ×16 water immersion objective (Nikon) was used. Each field-of-view was 118.45 μm × 62.93 μm with 1024 × 512 pixel. Acquisition took place at 30.3 frames-per-second with the ThorImageLS software (Thorlabs). In a second imaging session changes in sDarken fluorescence were recorded during electrical stimulation of the dorsal raphe in anesthetized mice. For this, mice were anesthetized with isoflurane and head-fixed under a ×20 water immersion objective (1.0 N/A, Zeiss). Time-lapses with a field-of-view sized 121.45 μm × 60.74 μm and 1024 × 512 pixel were acquired at 30.3 frames-per-second with the same custom-made

Thorlaps microscope. During each imaging session repeated electrical stimulations spaced by 20 s with a current of 750 μA at 50 Hz (100 μs pulses for 1 s) were delivered through the previously implanted bipolar twisted platinum electrodes using an isolated pulse stimulator (Model 2100, A-M Systems).

### Data analysis in vivo imaging

Data acquired during electrical stimulation and the reward-learning task was analyzed using a custom-made Python script. After detrending and filtering the data, the *z*-score was calculated with the following formula: $= \frac{x - \mu}{\sigma}$, whereby $x$ is the raw fluorescence, μ the mean and σ the standard deviation (Supplementary Fig. S17). For analysis of the electrical stimulation, the *z*-score values for single stimulations in each timelapse were averaged to generate an average for each animal. Furthermore, a grand average consisting of all averages for the single animals was generated. For analysis of the reward-related response, full-field fluorescence of one animal was used. The *z*-score fluorescence of a time window of −1.5 and +2.5 s (total 3.5 s) related to 11 reward events were extracted and averaged. For quantification, the median averaged baseline value was compared to the median averaged post-reward value. The baseline value was defined as the average of the 6 data points preceding the reward delivery (−0.2 to 0 s). The post-stimulation value was defined as the average constructed from the time window +0.5–+0.7 s. For analysis of the speed-modulation of the fluorescence, for each mouse all data points that related to a running speed of larger than 3 cm/s were averaged and compared to those that related to a running speed of smaller than 3 cm/s. This was done for all mice that showed extended periods of running (at least one running sequence longer than 10 s). Movement stops were then defined as sequences were the mouse runs for at least 4 s, stops running and then does not initiate another sequence of running for 4 s. The movement stops were identified for all mice that showed extended periods of running and then averaged. For the decay analysis the averaged fluorescence of the 4 sec after the stop of the movement was min-max-scaled using the MinMaxScaler of the Python Scikit-Learn package. A second-degree polynomial fit was performed on the curve and tau$_{1/2}$ was determined.

### Reporting summary

Further information on research design is available in the Nature Portfolio Reporting Summary linked to this article.

## Data availability

All data generated in this study are freely available upon request to the corresponding author. Source data are provided with this paper.

## Code availability

All custom codes used for analysis are available upon request.

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

## Acknowledgements

We thank Dr. Andy Tinker (UCL London, UK) who kindly provided HEK cells stably expressing GIRK1 and GIRK2 subunits. pAAV.CAG.SF-

iGluSnFR.A184V was a gift from L. Looger (Addgene plasmid #106199). We also thank Reinhard Seifert (Core Facility for Genetic Engineering, Research Center Caesar, Bonn) who kindly provided CNG channels. cpGFP from: pN1-GCamp6m XC (Addgene number #111543, Depositor: Xiaodong Liu). pAAV.Syn.NES.jRCaMP1a.W-PRE.SV40 was a gift from Douglas Kim & GENIE Project (Addgene plasmid # 100848). pAAV-Syn-ChrimsonR-tdT was a gift from Edward Boyden (Addgene plasmid # 5917). A.R. was supported by funding from the NRW-Rückkehrprogramm and the Deutsche Forschungsgemeinschaft (DFG RE 3101/3-1). O.A.M was funded by the Deutsche Forschungsgemeinschaft (DFG MA 4692/6-1 and - Projektnummer 122679504 - SFB 874). J.S.W. received funding for this project by the DFG (SPP 1926, WI4485-3/2, FOR2419, WI4485-2/2, and SFB935, B8). M.F. was supported by funding from the European Research Council (ERC-CoG MicroSynCom 865618).

## Author contributions

O.A.M. conceived and supervised the project. M.K. developed and optimized all sensors and variants. M.K. characterized together with L.W., S.B., K.R., J. Gerdey, J. Groß, P.G., K.C.C., and N.M. sDarken and variants in HEK cells and brain slices. Outside-out patch measurements were done by S.P., T.Z. and supervised by A.R. who also constructed the kinetic model. 2-Photon measurements in hippocampal slice culture were performed by M.P. and supervised by J.S.W. M. Müller and M. Mittag performed and analyzed in vivo 2-Photon experiments supervised by M.F. All authors contributed equally to the data analysis and writing of the manuscript.

## Funding

## Competing interests

The authors declare no competing interests.
