## [Peer Review File · Nature Communications]

REVIEWER COMMENTS

Reviewer #1 (Remarks to the Author):

Kubitschke et al. describe the engineering and characterization of a genetically encoded fluorescent biosensor for serotonin (5-HT) called sDarken. Inspired by the dLight indicator from the Tian lab, they replace the third intracellular loop of the human 5-HT_{1A} receptor with circularly permuted GFP. They optimize sensitivity by mutagenesis of the inter-domain linkers and make lower- and higher-affinity variants of sDarken by mutagenesis of the serotonin binding site and introduction of superfolder GFP mutations, respectively. The authors then characterize the performance of sDarken in cultured cells and demonstrate its use in brain slices and in vivo in mouse brain.

In the past couple years, several other GFP-based serotonin indicators have been published (see supplementary table 1), including with very similar combination of a serotonin-specific GPCR and cpGFP (GRAB5-HT). The authors likely started this work significantly before any of those other sensors were published. sDarken has smaller fluorescence change but faster kinetic response than GRAB5-HT. The primary differentiating feature of sDarken relative to previous sensors is that sDarken is a “negative-going” indicator (gets dimmer upon binding 5-HT). The authors don’t really focus on the negative-going aspect, which was likely just a chance consequence of exactly how they combined the serotonin receptor and cpGFP.

sDarken appears to be a relatively good serotonin indicator. The authors perform a thorough characterization of the brightness, sensitivity, specificity, photobleaching, localization and response kinetics in cultured cells. Demonstrations in vivo in mouse brain show that convincing signals can be observed with the indicator and that there was some correlation between sDarken signal and locomotion of the mouse. The authors highlight the use of 2P microscopy to image sDarken signals to achieve higher spatial resolution imaging of serotonin changes relative to the fiber photometry that was used in previous work. While they show one example (Fig. 6e) of spatially varying signals in vivo with sDarken, these data are not terribly convincing, and they do not make a strong case for the value of additional spatial information (although I personally believe that high spatial resolution will be important in the future).

In all, the new sDarken sensors presented are well-characterized and should be a valuable tool in a growing toolbox of neurotransmitter sensors for in vivo neuroscience.

Specific points that need to be addressed:

-I did not find amino acid sequences of the sDarken sensors anywhere in the manuscript. Annotated sequences should be added. I would also suggest to deposit plasmids for distribution at Addgene.

-Related to the sequences, the authors state “the third intracellular loop of the native 5-HT1A receptor was replaced by the circularly permuted form of GFP (cpGFP) also used in GCaMP6m”, but I’m not sure if this includes mutations to the cpGFP that were introduced during the engineering of GCaMP6m.

-In the description of library screening they state that they mutated two amino acids in linker 1 and two amino acids in linker 2 and analyzed 224 mutants. Were all four of the linker positions varied simultaneously? That would be a very large library (4^4). How did they decide upon the 224 variants that were assayed? Were they sequenced before assaying? I suggest to include more details of the protein engineering. Related – why does Fig. 1b have far fewer than 224 bars if that’s how many were screened?

-The authors state “We hypothesized that using a robust folding variant would increase the initial fluorescence and improve K_d values”. Why did they think that the superfolder GFP mutations would affect the ligand affinity in the GPCR domain? This does not seem like a logical expectation. If there are good reasons to expect this they should be stated more explicitly.

-“...the superfolder cpGFP improved the K_d twofold...” – ‘improved’ is subjective and depends on the application; change this to ‘decreased K_d ’ or ‘increased affinity’

-The authors state “We prepared acute brain slices of the IL of mice...”, but I can’t find the abbreviation IL spelled out anywhere.

-Figure 1d – perhaps add “washout” labels between the images

-Figure 2c – the Y-axis suggests that sDarken is positive-going? In general, the authors should try to use the same Y-axis units throughout to avoid confusion.

-Figures 2e and 2h are both showing photostability, but start at different places on the normalized Y-axis. Make these consistent.

-Figure 4 caption reads: “sDarken and its variants are suitable for 2-P applications and can report endogenous release of 5-HT”, but all panels in this figure only show exogenous application of serotonin, no endogenous release.

-Figure 5b – the Y-axis suggests $\Delta F/F$ is positive. Also looks like the two labels for the bars are switched.

-Figure 5c – the single-trial fluorescence traces shown appear to have 2-3x the response amplitude compared to the means for each stimulus shown in the bar graph inset. Please show more representative single-trial traces.

-Naming of the sDarken sensors is incorrect in supplementary table 1

-Change CAXX to CAAX

-The $\Delta F/F = (F_0 - F) / F_0$ definition provided in the methods (section Imaging in HEK cells) would only give positive values in the range 0-1 for signals that decrease from baseline. This does not match what is plotted in the corresponding figures.

-The word 'aus' is used in the Acknowledgements

Reviewer #2 (Remarks to the Author):

Summary

The neuromodulator serotonin (5-HT) is involved in a wide range of cognitive functions and is implicated in a number of psychiatric diseases. However, its functioning in the healthy brain remains elusive. 5-HT releasing neurons are located in small nuclei throughout the brainstem from which they send branching axons that innervate the entire brain through multiple (poorly resolved) pathways. Thus, tracking fluctuating 5-HT levels at specific brain regions is challenging, and conventional methods, such as microdialysis and FSCV, fall short of achieving this goal.

In this manuscript, the authors introduce a new family of genetically engineered 5-HT sensor proteins called sDarken. Focusing mainly on in vitro experiments and in vivo two-photon microscopy, they show that sDarken is a fast and sensitive sensor that may be used to measure cortical 5-HT levels at cellular and sub-second resolutions.

I found the molecular engineering and validation aspects of this work to be thorough and impressive. In a number of experiments, the authors clearly demonstrate that sDarken specifically responds to 5-HT and is sensitive enough to detect 5-HT concentrations in the sub-micromolar range. They further show that its expression does not harm neurons or interfere with endogenous second messenger signaling pathways. However, as detailed below, I was less convinced that sDarken provides reliable 5-HT readouts in vivo, particularly during behavior. This issue is concerning since, as the authors note, the major application of this tool would be in imaging experiments performed on awake behaving animals.

Major comments

1) Related to fig5: Electrical stimulation of the DRN led to a small and brief (even unnoticeable in one mouse) rise in 5-HT levels that was followed by a much larger and prolonged decrease. I found this result puzzling. While the authors claim that this may be the result of autoinhibition, I did not find support for this in their analysis or in their discussion regarding this observation. Their reference to Ren et al. does not support their claim and may even weaken it – the biphasic DR response to footshock there may be explained by recording average responses from a heterogeneous population of DR neurons.

2) Related to fig6: Here the authors use 2p microscopy to measure 5-HT levels in behaving mice. The image shown in panel e seems blurred and very different from the one shown in fig5 and in the paper by Wan et al., in which cell bodies appear as shadows over a fluorescent background. This raises the concern that movement artifacts, which are a well-known concern when imaging from behaving animals, affect, and may even dominate the reported signals. For example, in panel h 5-HT signals show very fast fluctuations that are remarkably similar to changes in running speed, and may reflect, at least in part, movement rather than genuine 5-HT changes.

In summary, I wish the authors would have applied some of the control and validation approaches that they used in their in vitro experiments also in the in vivo ones. For example, showing that local application of WAY-100635 abolishes reward and movement related activity (and perhaps also the dip that followed DRN stimulation) without affecting behavior, or that drugs that enhance and prolong 5-HT release (such as MDMA and SSRIs) boost 5-HT levels would have provided support for the applicability of sDarken for experiments in awake behaving mice.

3) Finally, a number of genetically encoded 5-HT sensors have been developed in the last couple of years. I would have liked to read a more detailed discussion regarding the relative advantages of sDarken compared to these alternative sensors.

Reviewer #3 (Remarks to the Author):

The authors have used straightforward methods to develop a GPCR-based serotonin sensor from the 5HT1A receptor, using the same principles previously used to develop the dLight dopamine sensor. This sensor (and the variants also produced here) appears to be an improvement over existing serotonin sensors both in terms of the fluorescence change and the affinity range covered. Controls were performed to demonstrate that the sensor, a modified GPCR, does not engage G-protein signaling. Expression and membrane targeting appear to be good, not only in HEK cells but also in organotypic slices, where strong responses are seen both to applied 5-HT and to local electrical stimulation. The sensor exhibits high specificity, rapid kinetics, and no desensitization.

Expression was also observed in vivo in a small number of mice, one of which appeared to show sensor responses that correlated with behavioral activity, and a few examples are shown to be correlated with ambulation. However, the data from animals are presented only in a heavily processed form (detrended, filtered, and z-scored). This makes it difficult to judge how usable the sensor is in vivo, particularly for quantitative characterization of 5-HT release. It would be helpful for the reviewers and for potential users of sDarken to see what the raw data look like (deltaF over initial baseline). Is detrending based on a linear decline over the whole experiment? Is this reasonable given that there are apparently local

increases in fluorescence that indicate decreased 5-HT, and thus chronic release, which could change over the course of the experiment? The data are also very anecdotal, with many panels in Fig. 6 based on results from only one mouse (though it's not completely clear which panels have more than one mouse). And do the traces in Fig 6j depict all of the movement stops during recording of these 3 mice, or just a selected subset? What were the fluorescence changes upon the start of movement?

It is important to perform pH characterization of the sensors. This is needed to determine if they respond to changes in pH alone (dose responses should be measured for a range of pH values, and results should not be separately normalized at each pH). Sensitivity to pH is very common for sensors with this cpGFP design, and changes in intracellular pH can certainly occur with neuronal activity.

The authors should at least discuss the possibilities that normal endogenous signaling could be affected by

- * buffering of 5-HT by the sensor
- * coassembly of the sensor with native 5-HTRs or competition for scaffolding proteins

The 5HT1A-specific antagonist, which blocks the response of the sensor to 5-HT, might be valuable in assessing the presence of baseline / background 5-HT release (as this would be seen as an apparent decrease in 5-HT [or increased fluorescence] from the initial level). The antagonist should be tested by itself on the sensor to see if it has inverse agonist effects on the fluorescence response (as may be suggested by the small overshoot seen in Supp. Fig. 2b). The antagonist might also be helpful in assessing any contribution of pH changes to the fluorescence signal, although the pH characterization would still be necessary as it would also indicate any pH sensitivity of the saturated sensor response, which may differ from the empty or antagonized sensor response.

Minor: Figure 4 does not show endogenous release of 5-HT as stated in the figure title. All of these responses are to bath-applied 5-HT.

We would like to thank the reviewers for their insightful comments. By revising the manuscript and adding more analysis and new experimental data and videos, we addressed the detail-oriented points raised by the reviewers as well as their concerns regarding the utility of *sDarken* for *in vivo* experimentation. Overall, this considerably improved the manuscript.

In the following section we will comment (in *italic*) on each of the reviewer suggestions and explain how we tried to integrate their suggestion and comments in the revised manuscript.

REVIEWER COMMENTS

Reviewer #1 (Remarks to the Author):

Kubitschke et al. describe the engineering and characterization of a genetically encoded fluorescent biosensor for serotonin (5-HT) called *sDarken*. Inspired by the *dLight* indicator from the Tian lab, they replace the third intracellular loop of the human 5-HT_{1A} receptor with circularly permuted GFP. They optimize sensitivity by mutagenesis of the inter-domain linkers and make lower- and higher-affinity variants of *sDarken* by mutagenesis of the serotonin binding site and introduction of superfolder GFP mutations, respectively. The authors then characterize the performance of *sDarken* in cultured cells and demonstrate its use in brain slices and *in vivo* in mouse brain.

In the past couple years, several other GFP-based serotonin indicators have been published (see supplementary table 1), including with very similar combination of a serotonin-specific GPCR and cpGFP (GRAB5-HT). The authors likely started this work significantly before any of those other sensors were published. *sDarken* has smaller fluorescence change but faster kinetic response than GRAB5-HT. The primary differentiating feature of *sDarken* relative to previous sensors is that *sDarken* is a “negative-going” indicator (gets dimmer upon binding 5-HT). The authors don’t really focus on the negative-going aspect, which was likely just a chance consequence of exactly how they combined the serotonin receptor and cpGFP.

sDarken appears to be a relatively good serotonin indicator. The authors perform a thorough characterization of the brightness, sensitivity, specificity, photobleaching, localization and response kinetics in cultured cells. Demonstrations *in vivo* in mouse brain show that convincing signals can be observed with the indicator and that there was some correlation between *sDarken* signal and locomotion of the mouse. The authors highlight the use of 2P microscopy to image *sDarken* signals to achieve higher spatial resolution imaging of serotonin changes relative to the fiber photometry that was used in previous work. While they show one example (Fig. 6e) of spatially varying signals *in vivo* with *sDarken*, these data are not terribly convincing, and they do not make a strong case for the value of additional spatial information (although I personally believe that high spatial resolution will be important in the future).

Reply: We thank the reviewer for the accurate evaluation. We added new figures, videos and data analysis to show the usefulness of *sDarken* *in vivo*. We agree that improved spatial resolution of *sDarken* would be an add. We therefore added further data showing cellular resolution imaging of *sDarken* in correlation to mouse behavior (Fig. 6 p-r). A subset of neurons in the PFC (Fig. 6 p, q, ROIs 1-4; **Supplementary Movie 14** displayed decreased fluorescence preceding a reward (licks), suggesting an anticipatory change in serotonin (Fig. 6 q, r). In contrast, neighboring neurons (ROIs 5-8) did not show fluorescence changes. These data show that *sDarken* can be used to record cellular serotonin events.

Figure 6 (p-r). (p) Average intensity projection of cellular layer 2/3 in PFC (Scale bar=25 μm). (q) Z-scored fluorescence traces of responding (black) and non-responding ROIs (grey) preceding reward-seeking behavior (licking (orange) and initiation of movement (light blue)). Responding ROIs (black) show short periods of decreased fluorescence in anticipation of reward-seeking behavior. (r) Average z-score of responding ROI 1-4 (black) and non-responding ROI 5-8 (grey).

We describe this observation in more depth in the discussion of our manuscript:

“Our experiments revealed a strong spatial variation in serotonin volume transmission, manifested in small localized hotspots, in which serotonin dynamics were evident anticipatory before the delivery of the reward (Li et al. 2016). Similar observations have been made for dopamine release in the cortex (Patriarchi et al. 2018). Future experiments will reveal if sDarken also allows for the visualization of serotonin release at dendrites, single spines or axons. Measurements in organotypic cultures and 2P measurements with cellular resolution in vivo suggest that also subcellular resolution, i.e. boutons and synapses, will be possible with sDarken.”

In all, the new sDarken sensors presented are well-characterized and should be a valuable tool in a growing toolbox of neurotransmitter sensors for in vivo neuroscience.

Specific points that need to be addressed:

-I did not find amino acid sequences of the sDarken sensors anywhere in the manuscript. Annotated sequences should be added.

Reply: We added full amino acid sequences of all sDarken sensors, please see new Supplemental file **Amino Acid Sequences**

L-sDarken

MDVLSPGQGNNNTSPPAPFETGGNTTGISDVTVSQVITSLLLGLTIFCAVLGNACVVAIALERSLQNVANYLIGSLAVTDLMVSVLVLPM AAL
YQVLNKWTLGQVTCDFIALNLVLCCTSSILHLCAIALDRYWAITDPIDYVKNRTPRRAAALISLTWLGFLISIPPMLGWRTPEDRSDPDACTISKD
HGYTIYSTFGAFYIPLLMLVLYGRIFRAARF**LSSYK**NVYIKADKQKNGIKANFKIRHNIEDGGVQLAYHYQQNTPIGDGPVLLPDNHYLSVQSKLS
KDPNEKRDHMLLEFVTAAGITLGMDELYKGGTGGSMVSKGEELFTGVVPILVELDGDVNGHKFSVSGEGEGDATYGKLT LKFICTTGKLPVP
WPTLVTTLTLYGVQCFSRYPDHMKQHDFFKSAMPEGYIQERTIFFKDDGNYKTRAEVKFEGDTLVNRIELKGI DFKEDGNILGHKLEYN**CFDQLR**
ERKTVKTLGIIMGTFILCWLPFFIVALVLPFCESSCHMPTLLGAIINWLGYSNLLNPVIYAYFNKDFQNAFKKIIKCKFCRQ*

sDarken

MDVLSPGQGNNNTSPPAPFETGGNTTGISDVTVSQVITSLLLGLTIFCAVLGNACVVAIALERSLQNVANYLIGSLAVTDLMVSVLVLPM AAL
YQVLNKWTLGQVTCDFIALDVLCTSSILHLCAIALDRYWAITDPIDYVKNRTPRRAAALISLTWLGFLISIPPMLGWRTPEDRSDPDACTISKD
HGYTIYSTFGAFYIPLLMLVLYGRIFRAARF**LSSYK**NVYIKADKQKNGIKANFKIRHNIEDGGVQLAYHYQQNTPIGDGPVLLPDNHYLSVQSKLS
KDPNEKRDHMLLEFVTAAGITLGMDELYKGGTGGSMVSKGEELFTGVVPILVELDGDVNGHKFSVSGEGEGDATYGKLT LKFICTTGKLPVP
WPTLVTTLTLYGVQCFSRYPDHMKQHDFFKSAMPEGYIQERTIFFKDDGNYKTRAEVKFEGDTLVNRIELKGI DFKEDGNILGHKLEYN**CFDQLR**
ERKTVKTLGIIMGTFILCWLPFFIVALVLPFCESSCHMPTLLGAIINWLGYSNLLNPVIYAYFNKDFQNAFKKIIKCKFCRQ*

H-sDarken

MDVLSPGQGNNNTSPPAPFETGGNTTGISDVTVSQVITSLLLGLTIFCAVLGNACVVAIALERSLQNVANYLIGSLAVTDLMVSVLVLPM AAL
YQVLNKWTLGQVTCDFIALDVLCTSSILHLCAIALDRYWAITDPIDYVKNRTPRRAAALISLTWLGFLISIPPMLGWRTPEDRSDPDACTISKD
HGYTIYSTFGAFYIPLLMLVLYGRIFRAARF**LSSYK**NVYITADKQKNGIKANFKIRHNVEDGGSVQLADHYQQNTPIGDGPVLLPDNHYSTQSVL
SKDPNEKRDHMLLEFVTAAGITLGMDELYKGGTGGSMVSKGEELFTGVVPILVELDGDVNGHKFSVRGEGEGDATNGKLT LKFICTTGKLPV
PWPTLVTTLTLYGVQCFSRYPDHMKRHDFFKSAMPEGYVQERTISFKDDGTYKTRAEVKFEGDTLVNRIELKGI DFKEDGNILGHKLEYN**CFDQLR**
RERKTVKTLGIIMGTFILCWLPFFIVALVLPFCESSCHMPTLLGAIINWLGYSNLLNPVIYAYFNKDFQNAFKKIIKCKFCRQ*

I would also suggest to deposit plasmids for distribution at Addgene.

Reply: We already deposited our sDarken expression plasmids (pAAV-Syn-sDarken/- H-sDarken/-L-sDarken and pN1-CMV sDarken/- H-sDarken/-L-sDarken) to Addgene. As soon as the paper will be published, they will be available through Addgene.

-Related to the sequences, the authors state “the third intracellular loop of the native 5-HT1A receptor was replaced by the circularly permuted form of GFP (cpGFP) also used in GCaMP6m”, but I’m not sure if this includes mutations to the cpGFP that were introduced during the engineering of GCaMP6m.

*Reply: We used the cpGFP that is part of GCaMP6m,” this means also mutations that were introduced during the engineering of GCaMP6m are included (AS 65, 75, 87, 92, 115, 118, 250). For further details please also refer to the new supplemental file **Amino Acid Sequences***

-In the description of library screening they state that they mutated two amino acids in linker 1 and two amino acids in linker 2 and analyzed 224 mutants. Were all four of the linker positions varied simultaneously? That would be a very large library ($4^4=256$). How did they decide upon the 224 variants that were assayed? Were they sequenced before assaying? I suggest to include more details of the protein engineering. Related – why does Fig. 1b have far fewer than 224 bars if that’s how many were screened?

*Reply: Please see new **Supplementary Figure S2** and the detailed description to this figure.*

Supplementary figure S2: Library design. In total 224 mutants were expressed and analyzed in HEK cells. In the first round of mutations, in which we followed the design strategies of Patriarichi et al. 2018, 58 mutants were designed according to the replacement strategy used for the 5-HT2A in Patriarichi et al. 2018 and 12 mutants were generated fitting based on dLight. Out of these 58 analyzed mutants only 41 showed fluorescence in the membrane and functionality. The 12 mutants according to dLight were not different in functionality or expression. No obvious pattern for functionality was evident except a favor for aromatic amino acids in position 2 and 3 of the C-terminal Linker. In a second round of mutations several strategies were applied in parallel to further increase the number of functional mutants: First, position 4 and 5 in the N-terminal linker 1 were again randomly mutated, while retaining an aromatic amino (F/W) acid in the C-terminal linker at position 2 and 3. 70 of these mutants were randomly chosen and further analyzed. In parallel, we selected the two mutants with an increase in fluorescence and mutated them further in different linker positions. 44 of these were further analyzed but revealed no further improvement in desired properties. In addition, according to Patriarichi et al. 2018 we changed the insertion site of cpGFP and analyzed another 20 mutants, in which the position of the cpGFP was varied. In a third round of mutations we started with the most promising mutant so far (M34) and mutated either position 1, 2, 3, of the N-terminal Linker or position 3, 4, 5 of the C-terminal linker randomly. None of the 20 additional mutants showed any improvement over the original variant (M34). As mutant M34 already showed a high and stable brightness, a large change in signal amplitude upon application of serotonin and superior expression in the membrane, we decided to characterize M34, from now on termed *sDarken*, further.

-The authors state “We hypothesized that using a robust folding variant would increase the initial fluorescence and improve K_d values”. Why did they think that the superfolder GFP mutations would affect the ligand affinity in the GPCR domain? This does not seem like a logical expectation. If there are good reasons to expect this they should be stated more explicitly.

Reply: According to several other publications (Muto et al. 2011, Qin et al. 2016 and St-Pierre et al. 2014) the exchange of cpGFP with the superfolder variant changed dynamic range, brightness, photostability and sensitivity of GCaMP, a Zn^{2+} sensor and a voltage sensor. We added a short explanation in the manuscript:

“ In different biosensors, i.e. GCaMP, Zn^{2+} and voltage sensor) an exchange of cpGFP with the superfolder variant improved sensitivity, dynamic range, brightness and photostability {Muto:2011cp, Qin:2016ky, StPierre:2014db. Hence we hypothesized that using a superfolder variant of cpGFP would increase the initial fluorescence and potentially decrease K_d values ^{23,24}”

-“...the superfolder cpGFP improved the K_d twofold...” – ‘improved’ is subjective and depends on the application; change this to ‘decreased K_d ’ or ‘increased affinity’

Reply: We agree and we changed the wording as suggested:

*” Indeed, substitution of cpGFP with the superfolder cpGFP increased the affinity twofold to 57 nM (57.3 ± 13.9 nM) in comparison to the original sDarken (**Fig. 2c**), whereas the signal amplitude remained unchanged (**Fig. 2c**).”*

-The authors state “We prepared acute brain slices of the IL of mice...”, but I can’t find the abbreviation IL spelled out anywhere.

Reply: Please apologize, that we have not spelled out IL anywhere before. We now added the description infralimbic (IL).

-Figure 1d – perhaps add “washout” labels between the images

*Reply: As suggested, we added “washout” labels in **Figure 1d**.*

(d) i Representative image sequence of repetitive 5-HT application via wash in. ii Example fluorescence measurement in HEK cells, repetitive application of 5 μ M 5-HT in 9 example cells. Application of 5-HT leads to a reversible reduction of fluorescence. No baseline corrections were performed. iii Heatmap of all analyzed ROIs.

-Figure 2c – the Y-axis suggests that sDarken is positive-going? In general, the authors should try to use the same Y-axis units throughout to avoid confusion.

Reply: Thank you for this important remark. We apologize for the inconsistent explanation and visualization. To visualize the decrease in fluorescence of sDarken due to binding of serotonin we show raw traces with the observed decrease (negative change) in fluorescence. In quantifications, however, we now consistently report the typical fluorescence decrease of sDarken as a “positive” bar, labeled “-ΔF/F”. This allows for a more intuitive comparison to existing positive-going sensors and assessment of dose-response curves, agonist specificity and so on.

-Figures 2e and 2h are both showing photostability, but start at different places on the normalized Y-axis. Make these consistent.

Reply: We corrected the labelling in **Figure 2e** and **h** accordingly.

-Figure 4 caption reads: “sDarken and its variants are suitable for 2-P applications and can report endogenous release of 5-HT”, but all panels in this figure only show exogenous application of serotonin, no endogenous release.

Reply: We apologize for this misleading legend, we corrected the label, please see **Figure 4**:

“Figure 4: sDarken and its variants are suitable for 2-P applications.”

-Figure 5b – the Y-axis suggests $\Delta F/F$ is positive. Also looks like the two labels for the bars are switched.

Reply: We apologize for the wrong labelling, **Figure 5b** is corrected accordingly.

Figure 5: Endogenous release of serotonin in brain slices and in the anesthetized mice. (b) Quantification of fluorescence decrease after puff application from data in a, *** $p < 0.001$.

-Figure 5c – the single-trial fluorescence traces shown appear to have 2-3x the response amplitude compared to the means for each stimulus shown in the bar graph inset. Please show more representative single-trial traces.

Reply: We show now more representative single traces, please see **Figure 5c**

Figure 5: Endogenous release of serotonin in brain slices and in the anesthetized mice (c) A bipolar stimulation electrode was positioned in the field of view and electrical pulses @40 Hz were delivered (0.2ms, 750 μ A). Single traces for electrical stimulation with different number of pulses inset Quantification of fluorescence responses to the indicated pulse number.

-Naming of the sDarken sensors is incorrect in supplementary table 1

Reply: We apologize for the incorrect naming in supplementary table1, we changed the naming as suggested.

Table 1: Comparison of genetically encoded serotonin sensors									
Sensor	($\Delta F/F_0$) % change						Kd	τ_{ON} (ms)	τ_{OFF} (ms)
	10 nM	100 nM	1.6 μ M	3.2 μ M	100 μ M	1 mM			
L-sDarken	-	-	-	- 20	- 48	-73	145 μ M	95	156
sDarken	- 14	- 33	- 58	- 60	-40	Sat.	127 nM	43	323
H-sDarken	- 8	- 51	- 62	- 67	Sat.	Sat.	57 nM	57	324
GRAB_{5-HT}¹	70	240	280	Sat	Sat	Sat	22 nM	200	3100
iSero²	15	20	30	75	750	Sat	390 μ M	0.5-10 (fast) 5-18 (slow)	4 (fast) 150 (slow)
PsychLight³	25	55	80	Sat	Sat	Sat	26.3 nM		997 (fast) 3998 (slow)

-Change CAXX to CAAX

Reply: We corrected the misspelling of CAAX in the entire manuscript.

-The $\Delta F/F = (F_0 - F) / F_0$ definition provided in the methods (section Imaging in HEK cells) would only give positive values in the range 0-1 for signals that decrease from baseline. This does not match what is plotted in the corresponding figures.

Reply: To visualize the decrease in fluorescence of sDarken due to binding of serotonin we show fluorescence traces with % change in fluorescence. For a better comparison to existing positive going sensors, we report the typical decrease of sDarken in positive direction (but labeled -dF/F for Kd plots and agonist specificity (see above).

-The word 'aus' is used in the Acknowledgements

Reply: We changed the german word "aus" to "from"

Reviewer #2 (Remarks to the Author):

Summary

The neuromodulator serotonin (5-HT) is involved in a wide range of cognitive functions and is implicated in a number of psychiatric diseases. However, its functioning in the healthy brain remains elusive. 5-HT releasing neurons are located in small nuclei throughout the brainstem from which they send branching axons that innervate the entire brain through multiple (poorly resolved) pathways. Thus, tracking fluctuating 5-HT levels at specific brain regions is challenging, and conventional methods, such as microdialysis and FSCV, fall short of achieving this goal.

In this manuscript, the authors introduce a new family of genetically engineered 5-HT sensor proteins called sDarken. Focusing mainly on in vitro experiments and in vivo two-photon microscopy, they show that sDarken is a fast and sensitive sensor that may be used to measure cortical 5-HT levels at cellular and sub-second resolutions.

I found the molecular engineering and validation aspects of this work to be thorough and impressive. In a number of experiments, the authors clearly demonstrate that sDarken specifically responds to 5-HT and is sensitive enough to detect 5-HT concentrations in the sub-micromolar range. They further show that its expression does not harm neurons or interfere with endogenous second messenger signaling pathways. However, as detailed below, I was less convinced that sDarken provides reliable 5-HT readouts in vivo, particularly during behavior. This issue is concerning since, as the authors note, the major application of this tool would be in imaging experiments performed on awake behaving animals.

Reply: We thank the reviewer for this detailed and overall positive assessment of our work.

Major comments

1) Related to fig5: Electrical stimulation of the DRN led to a small and brief (even unnoticeable in one mouse) rise in 5-HT levels that was followed by a much larger and prolonged decrease. I found this result puzzling. While the authors claim that this may be the result of autoinhibition, I did not find support for this in their analysis or in their discussion regarding this observation. Their reference to Ren et al. does not support their claim and may even weaken it – the biphasic DR response to footshock there may be explained by recording average responses from a heterogeneous population of DR neurons.

Reply: We agree with the reviewer and we were at first also surprised to find these response kinetics upon electrical stimulation of DR. However, this response was detected very reliable and is already with precedence in literature. We also agree that there are other possibilities and network effects that could contribute to the observed kinetics. However, we think it is out of the scope of this manuscript to identify them. To incorporate these different possibilities, we include them in the discussion and elaborate the topic more carefully in the discussion now. In addition, a very recent publication from Paquelet et al. 2022. shows that DR neurons have a mixed selectivity and heterogeneous projections, adding further support for our notion:

“The same biphasic phenomenon was observed in footshock experiments performed with psychLight2 (another GPCR based serotonin sensor) in the DR and in the prefrontal cortex (Dong et al. 2021). Fiber photometry recordings of serotonergic neurons in the DR, when studied as one population, display also biphasic responses to footshock, however, these response properties were thought to originate from anatomically distinct subpopulations (Ren et al.2018). Recent miniscope imaging with single cell resolution, has now impressively shown, that different response types (activation or inhibition) to emotional stimuli, such as reward or footshock, are even present in anatomically defined projections {Paquelet:2022fu}. This heterogeneity could be one explanation to our observed biphasic response in the PFC.

Another explanation might be that the biphasic response is a circuit phenomenon mediated by either autoinhibition within the DR or by top-down projections from the PFC to the DR (activation of a negative feedback mechanism), as biphasic serotonin dynamics in the PFC were only apparent in vivo, whereas upon electrical stimulation in vitro only an increase in serotonin levels was observed. In addition, due to the close proximity to the median raphe (MR) we cannot exclude, that the electrical stimulation activated also MR serotonergic neurons that might contribute to the biphasic response in the PFC {Muzerelle:2016dw}. Besides, electrical stimulation might cause an excessive release of serotonin followed by a subsequent depletion of serotonin stores at the synapse, causing a prolonged decrease in serotonergic tone.”

2) Related to fig6: Here the authors use 2p microscopy to measure 5-HT levels in behaving mice. The image shown in panel e seems blurred and very different from the one shown in fig5 and in the paper by Wan et al., in which cell bodies appear as shadows over a fluorescent background. This raises the concern that movement artifacts, which are a well-known concern when imaging from behaving animals, affect, and may even dominate the reported signals. For example, in panel h 5-HT signals show very fast fluctuations that are remarkably similar to changes in running speed, and may reflect, at least in part, movement rather than genuine 5-HT changes.

Reply: We thank the reviewer for this precise and valuable observation. Indeed, the blurry impression with no cell bodies of **Fig. 6e** (now **Fig. 6h**) in comparison to **Fig. 5f** results from the fact that **Fig. 6e** recordings were in cortical layer 1 in comparison to layer 2/3 for **Fig. 5f**. Since there are no neuronal cell bodies expressing sDarken in layer 1, but rather dendrites, **Fig. 6e** does not contain dark nuclei of neurons. To clarify in which layer we imaged, we included a schematic similar to **Fig. 5e** (**Fig. 6e**) and indicated the layer in the figure. Furthermore, we provide fluorescence images of the injection sites of sDarken in the PFC (**Fig. 6f**) as well as exemplary field-of-views for two-photon imaging in layer 1 and layer 2/3 **Fig. 6g**).

Figure 6 (e-g). (e) Schematic representation of cranial window above PFC. (f) Histological images of cortical AAV-Syn-sDarken expression. Left image: Overview of ipsilateral hemisphere (Scale bar=1mm). Right image: Blow up of layer 1 and layer 2/3 (Scale bar=100μm). (g) Two-photon imaging field-of-view of AA-Syn-sDarken expression in PFC layer 1 (top) and layer 2/3 (bottom). (Scale bar=50μm).

Concerning movement artifacts, it is definitely true that they are affecting the fluorescence signal and that this becomes especially prominent during periods of running. This is why - in order to validate that there is fluorescence dynamics unrelated to shifts in focal plane - we specifically analyzed the fluorescence decay after the animals stopped running. Within the four seconds after stop of movement, we observed a slow decay of fluorescence down to stable baseline levels (**Fig. 6 n, o**). These changes suggest a switch between low levels of serotonin during running and higher levels during rest. In order to explore serotonergic signaling further, in the updated manuscript we added an analysis (**Fig. 6 p-r**) that focuses on cell bodies in layer 2/3 of mPFC. The appearance of cellular structures in this field-of-view much more resembles the presentation in Wan et al. 2021 (**Fig. 6 p**). Here, in one mouse, in three instances we found that local fluorescence events seem to anticipate reward-seeking behavior.

Figure 6: Two-photon *in-vivo* imaging of sDarken in awake animal. (m) Grey traces show single fluorescence traces ($n=16$) related to movement starts in three mice. Black trace shows averaged trace. (n) Grey traces show single fluorescence traces ($n=18$) related to movement stops. Black trace shows averaged trace. (o) Black trace shows min-max-scaled averaged fluorescence decay after movement stops. Grey line indicates 2nd-degree polynomial fit of fluorescence decay ($\tau_{off}=1.24$ sec). (p) Average intensity projection of cellular layer 2/3 in PFC (Scale bar= $25 \mu\text{m}$). (q) Z-scored fluorescence traces of responding (black) and non-responding ROIs (grey) preceding reward-seeking behavior (licking (orange) and initiation of movement (light blue)). Responding ROIs (black) show short periods of decreased fluorescence in anticipation of reward-seeking behavior. (r) Average z-score of responding ROI 1-4 (black) and non-responding ROI 5-8 (grey).

Furthermore, we added movies (**Supplementary Movie 13-14**) to support our figures and to showcase behavior-related as well as electrical stimulation-induced fluorescence changes. Movie 13 (**Supplementary Movie 13-14**) displays decreased fluorescence in 4 ROIs preceding a reward (licks), suggesting an anticipatory change in serotonin (**Fig. 6 q, r**). The second video (**Supplementary Movie 14**) shows the average of 9×20 second-events of sDarken fluorescence change in PFC layer 2/3 upon electrical stimulations in DR. The movie relates to the example shown in Fig. 5 f. In this movie it can be observed, that after electrical stimulation in DR there first is a drop of sDarken fluorescence which then is followed by an increase in fluorescence.

In summary, I wish the authors would have applied some of the control and validation approaches that they used in their *in vitro* experiments also in the *in vivo* ones. For example, showing that local application of WAY-100635 abolishes reward and movement related activity (and perhaps also the dip that followed DRN stimulation) without affecting behavior, or that drugs that enhance and prolong 5-HT release (such as MDMA and SSRIs) boost 5-HT levels would have provided support for the applicability of sDarken for experiments in awake behaving mice.

Reply: We thank the reviewer for this suggestion as these experiments would strengthen the characterization of sDarken *in vivo*. Nevertheless, we are unable to provide these experiments in a reasonable amount of time. A new animal experimental license would be necessary for this, which will not be approved before a period of at least 6-12 months. We hope that the new added data, figures and videos could convince you that sDarken is applicable *in vivo*. Also, in regards of the 3 R principles we think these additional experiments would be out of scope of the manuscript. In any case, we believe that the new added data, figures and videos demonstrate convincingly that sDarken is suitable for *in vivo* studies. We adapted our statement in the discussion accordingly:

“Our experiments also show the potential of sDarken in 2P imaging and its superior membrane expression, brightness and stability. Kinetic properties, in combination with superior brightness and photostability will have advantages in resolving subcellular structures. In the future, specific targeting to synapses could allow for high resolution imaging of serotonin release even at synaptic terminals.”

3) Finally, a number of genetically encoded 5-HT sensors have been developed in the last couple of years. I would have liked to read a more detailed discussion regarding the relative advantages of *sDarken* compared to these alternative sensors.

Reply: Thank you for this valuable comment, we included a more detailed discussion regarding the advantages of *sDarken* and comparison to existing sensors, please see the discussion:

“In comparison to other recently developed 5-HT sensors (GRAB_{5-HT}, PsychLight2 and iSeroSnFR)^{5,9,13,18,34}, which either have fast kinetics or high affinity, *sDarken* shows high sensitivity combined with relatively fast kinetics (**Supplementary Table1**). Although GRAB_{5-HT} and PsychLight2 use a similar design strategy, i.e., introduction or replacement of IL3, the use of different serotonin receptors as sensing moiety, linkers and insertion sites yielded sensors with distinct properties. For instance, in comparison to PsychLight and *sDarken*, GRAB sensors do not substitute ICL3 completely, which might lead to residual activation of downstream signaling⁵. This might necessitate the control for undesirable side effects in some research applications. *sDarken* is the only serotonin sensor, which is bright in the unbound state and diminishes its fluorescence upon binding of serotonin. In comparison to conventional neurotransmitter sensors *sDarken* has a brightness that is comparable to membrane expression of eGFP. This superior brightness will be favorable for imaging in general. In addition, *H-sDarken* has a three times higher brightness in comparison to eGFP, well in accordance with properties of superfolder variants²³. Our experiments also show the potential of *sDarken* in 2P imaging and its superior membrane expression, brightness and stability. Kinetic properties, in combination with superior brightness and photostability will have advantages in resolving subcellular structures. (Pulin et al.2022). In the future, specific targeting to synapses could allow for high resolution imaging of serotonin release even at synaptic terminals.

Existing serotonin sensors are either based on 5-HT₂ receptors {Dong:2021ge} (Dong et al. 2021, Wang et al. 2021) or on a mutated periplasmic binding protein from *E.coli*. (Unger et al. 2021), whereas *sDarken* is based on the 5-HT_{1A} receptor. 5-HT_{1A} receptors are a major target for anxiolytic, antidepressant and antipsychotic medications. *sDarken* could be used in a similar manner as PsychLight {Dong:2021ge} to identify new compounds capable of binding to 5-HT_{1A} receptors, that might be useful in finding new treatments strategies for psychiatric diseases.”

Reviewer #3 (Remarks to the Author):

The authors have used straightforward methods to develop a GPCR-based serotonin sensor from the 5HT1A receptor, using the same principles previously used to develop the dLight dopamine sensor. This sensor (and the variants also produced here) appears to be an improvement over existing serotonin sensors both in terms of the fluorescence change and the affinity range covered. Controls were performed to demonstrate that the sensor, a modified GPCR, does not engage G-protein signaling. Expression and membrane targeting appear to be good, not only in HEK cells but also in organotypic slices, where strong responses are seen both to applied 5-HT and to local electrical stimulation. The sensor exhibits high specificity, rapid kinetics, and no desensitization.

Reply: We thank the reviewer for this positive assessment of our sensor design and characterization in vitro and in situ.

Expression was also observed in vivo in a small number of mice, one of which appeared to show sensor responses that correlated with behavioral activity, and a few examples are shown to be correlated with ambulation. However, the data from animals are presented only in a heavily processed form (detrended, filtered, and z-scored). This makes it difficult to judge how usable the sensor is in vivo, particularly for quantitative characterization of 5-HT release. It would be helpful for the reviewers and for potential users of sDarken to see what the raw data look like (deltaF over initial baseline). Is detrending based on a linear decline over the whole experiment? Is this reasonable given that there are apparently local increases in fluorescence that indicate decreased 5-HT, and thus chronic release, which could change over the course of the experiment? The data are also very anecdotal, with many panels in Fig. 6 based on results from only one mouse (though it's not completely clear which panels have more than one mouse). And do the traces in Fig 6j depict all of the movement stops during recording of these 3 mice, or just a selected subset? What were the fluorescence changes upon the start of movement?

Reply: We thank the reviewer for these valuable comments. It is true that the data was processed to a certain extent, but all features were also clearly visible in the raw data. During recording, we experienced a constant linear decline of fluorescence likely due to bleaching of sDarken. In **Supplementary figure 17** we now show an example of how the different processing steps influence the raw data. **Supplementary figure 17** (a) shows an exemplary raw fluorescence trace (blue) together with the linear regression (orange). In (b) one can see the detrended fluorescence trace. **Supplementary figure 17** (c) depicts the z-scored fluorescence and (d) provides further information about behavioral readouts like running speed (green) and licking (red). As you can appreciate, the overall features of the data remain unaffected.

Supplementary Figure S17. (a) Raw fluorescence data of a full field ROI. (b) Detrended raw fluorescence based on linear regression. (c) Z-scored fluorescence. (d) Z-scored fluorescence (blue) plotted together with running speed (green) and licking (red).

The processing of the data was necessary, as the linear rundown (bleaching) in the data did not correlate with any physiological signal and the removal allowed for analysis of other serotonin dynamics related to the animal's behavior. To clarify the effect of data processing we included **Fig. S17** as a **Supplementary figure S17**. Furthermore, we included exemplary movies of sDarken fluorescence changes (**Supl. Movie 13, 14**)

It is true that during the analysis of different aspects of sDarken dynamics in some cases we were relying on data obtained from only one mouse. This is specifically the case for the analysis of the reward-related signal (**Fig. 6 i, j**) and the description of a local fluorescence event (**Fig. 6 h**). To make this clearer, we have additionally stressed the number of mice used for each analysis in the legend of Fig. 6.

“Figure 6. Two-photon in-vivo imaging of sDarken in awake animal. (a) Time schedule of experiment. (b) Schematic depiction of linear treadmill with local spatial textures. Reward positions are indicated by orange box. (c) Representational image of head-fixed animal running on linear treadmill. (d) Summed number of licks over spatial bins (grey bars) of mice (n=2) that successfully learned the task. Heatmap indicates averaged velocity over spatial bins. (e) Schematic representation of cranial window above PFC. (f) Histological images of cortical AAV-Syn-sDarken expression. Left image: Overview of ipsilateral hemisphere (Scale bar=1mm). Right image: Blow up of layer 1 and layer 2/3 (Scale bar=100µm). (g) Two-photon imaging field-of-view of AA-Syn-sDarken expression in PFC layer 1 (top) and layer 2/3 (bottom). (Scale bar=50µm). (h) Large image shows average intensity projection of sDarken expression in dendritic layer in PFC (Scale bar=100 µm). Small images show 16-colors intensity projection of ROI1 at different time intervals (TP0 – TP2). Black trace shows fluorescence depicted as z-score, blue trace shows velocity of mouse. (i) Three exemplary traces of full field fluorescence changes in response to pump activation/reward delivery. Right trace shows average of 11 reward events in one animal. (j) Quantification of average z-score at (median=2.76, n=11 reward events in one animal) and post-reward (median=2.10, n=11). $P < 0.01$, Wilcoxon matched-pairs signed ranks test. (k) Black trace shows full-field sDarken fluorescence, blue trace indicates velocity of mouse. (l) Quantification of average fluorescence during periods of running (speed > 3 cm/sec) and periods of rest (speed < 3 cm/sec) in mice that showed extended periods of running (running: mean=1.94; rest: mean=0.42; n=3 mice). $P < 0.05$, Paired t-test. (m) Grey traces show single fluorescence traces (n=16) related to movement starts in three mice. Black trace shows averaged trace. (n) Grey traces show single fluorescence traces (n=18) related to movement stops in three mice. Black trace shows averaged trace. (o) Black trace shows min-max-scaled averaged fluorescence decay after movement stops. Grey line indicates 2nd-degree polynomial fit of fluorescence decay ($\tau_{off}=1.24$ sec). (p) Average intensity projection of cellular layer 2/3 in PFC (Scale bar=25 µm). (q) Z-scored fluorescence traces of responding (black) and non-responding ROIs (grey) preceding reward-seeking behavior (licking (orange) and initiation of movement (light blue)). Responding ROIs (black) show short periods of decreased fluorescence in anticipation of reward-seeking behavior. (r) Average z-score of responding ROI 1-4 (black) and non-responding ROI 5-8 (grey).”

The reason why we had to rely on single animals for some of the analysis is that local events of serotonin release seem to be rather sparse. For future experiments, it seems therefore advisable to express a structural marker in serotonergic fibers so that sources of potential serotonin release can be easier located.

For the analysis of fluorescence dynamics related to movement stops we looked at all movement stops that were followed by a period of rest for at least four seconds. This limitation was necessary because it allowed us to explore the fluorescence decay without the influence of movement-induced artifacts. In total we observed 18 movement stops in three animals. In **Fig. 6 m**, we furthermore show the fluorescence change upon onset of movement. Here we selected only those movement starts that were preceded by a period of rest for at least three seconds. In total we observed 16 movement starts in the same three animals.

Figure 6. (m) Grey traces show single fluorescence traces (n=16) related to movement starts in three mice. Black trace shows averaged trace.

The data show a steep increase of running velocity upon movement initiation (blue trace). In contrast, the fluorescence (black trace) rises much slower than it would be expected if it was related to movement artifacts. We are therefore confident that despite the existence of movement artifacts sDarken reliably reports the changes of serotonin levels in vivo.

It is important to perform pH characterization of the sensors. This is needed to determine if they respond to changes in pH alone (dose responses should be measured for a range of pH values, and results should not be separately normalized at each pH). Sensitivity to pH is very common for sensors with this cpGFP design, and changes in intracellular pH can certainly occur with neuronal activity.

Reply: We performed additional pH characterization in HEK cells of the sensor, please see **Supplementary figure S4** sDarken did not show any noteworthy responses to changes in pH:

Supplementary figure S4: Influence of different pH values on sDarken fluorescence. a Measurement of fluorescence stability over time during different pH values, either without 5-HT (i) or with 5-HT (100 μ M) (ii). No apparent effect of the different pH values could be spotted. Individual colored lines represent the mean over time. b) Fluorescence changes due to 5-HT stimulation (post-pre; 5 frame average before and after 5-HT wash in) during different pH values. No significant differences could be detected between the different pH values (Welch's ANOVA, Dunnett's Post Hoc Test). c) Mean time courses \pm standard deviation of the data plotted in b.

As mentioned from the reviewer cpGFP is known for being sensitive to pH changes, but as with all other GPCR based biosensors, such as PsychLight and GRAB the cpGFP is located on the cytosolic side of the cell: Intracellular changes of pH values are expected to be only marginal during action potential firing (Pressler et al 1988 pH changes about ± 0.04 during AP) and are not expected to influence the fluorescence of sDarken only in a marginal way.

The authors should at least discuss the possibilities that normal endogenous signaling could be affected by
* buffering of 5-HT by the sensor
* coassembly of the sensor with native 5-HTRs or competition for scaffolding proteins

Reply: Thank you for this insightful comment, we have now included a discussion on the above-mentioned points:

“Obviously, the expression of sensor proteins for neurotransmitters can distort their diffusion/reuptake dynamics and the signals they elicit by buffering the analyte (see e.g. Armbruster et al. 2020). These effects can be circumvented by choosing a variant with an ideal affinity range (sDarken, L-sDarken, or H-sDarken) and by keeping the expression levels as low as possible. The strong fluorescence and large signal changes of the sDarken family is particularly helpful in that regard. At the same time, low expression levels will reduce possible assembly with native 5-HT_{1A} or other receptors into homo- or heterodimers, respectively, or interactions with other scaffolding proteins. Other biosensors, that utilize the same design principle as sDarken, did not show any significant effects on endogenous 5-HT receptor signaling Wan et al 2021.

The 5HT_{1A}-specific antagonist, which blocks the response of the sensor to 5-HT, might be valuable in assessing the presence of baseline / background 5-HT release (as this would be seen as an apparent decrease in 5-HT [or increased fluorescence] from the initial level). The antagonist should be tested by itself on the sensor to see if it has inverse agonist effects on the fluorescence response (as may be suggested by the small overshoot seen in Supp. Fig. 2b). The antagonist might also be helpful in assessing any contribution of pH changes to the fluorescence signal, although the pH characterization would still be necessary as it would also indicate any pH sensitivity of the saturated sensor response, which may differ from the empty or antagonized sensor response.

Reply: Thank you for this very important note. This is an interesting possibility to investigate baseline levels of 5-HT. As can be seen from the new included Supplementary Fig.2 c WAY indeed behaves as an inverse agonist (as suspected by the reviewer) and increases the initial fluorescence of sDarken further.

Supplementary figure S3: Response of sDarken to the selective 5-HT_{1A} antagonist WAY 100635. c) Application of WAY 100635 only at frame 30-40, n=23, mean ± SEM.

In future experiments this property could be used to assess baseline levels of serotonin, for example as mediated by the tonic activity of serotonergic neurons during wakefulness. Unfortunately, these elaborated experiments are out of the scope of this manuscript

Minor: Figure 4 does not show endogenous release of 5-HT as stated in the figure title. All of these responses are to bath-applied 5-HT.

Reply: We apologize for this misleading title, we corrected it, please see **Figure 4**.

“Figure 4: sDarken and its variants are suitable for 2-P applications.”

REVIEWER COMMENTS

Reviewer #1 (Remarks to the Author):

Generally, the authors addressed our concerns/comments and improved their manuscript. Although I appreciate the addition of Supp fig 2 to better explain the protein engineering, the graphs within that figure are confusing. They have no units on the y-axis, there seem to be many fewer bars than the number of clones analyzed, and the top two graphs are exactly the same even though these should represent different libraries.

Reviewer #2 (Remarks to the Author):

The authors have adequately addressed my comments and concerns.

Reviewer #3 (Remarks to the Author):

Many clarifications have been made in the revised manuscript, but I have two important remaining concerns.

1. The concerns about pH sensitivity have not been adequately addressed. The experiments of Supplementary Figure S4 are not correctly designed to reveal pH effects, either on the baseline fluorescence of the sensor or on the dose-response curve (i.e. on the $K_{apparent}$ for 5-HT). As requested in the original review, the authors should measure the effects of pH on samples that are NOT separately normalized (in the current Supplementary Fig 4C, each sample has already been baseline subtracted separately and normalized to its initial fluorescence at its particular pH). Without this information, it is impossible to know whether small pH changes can mimic the effect of 5HT.

Ideally the fluorescence should be observed during a pH change. Additionally, different doses of 5-HT should be used to establish whether pH changes the apparent affinity of the sensor for the ligand.

2. The inverse agonist behavior of the WAY compound (shown in the authors' response and in Supplementary Figure S3) is interesting - but it actually impairs the ability to use WAY to investigate baseline levels of 5HT. As shown in the figure, even with zero starting [5HT], the WAY compound increases fluorescence and decreases the "apparent 5HT concentration", and this could be incorrectly interpreted as a non-zero concentration of 5HT before the WAY is added. The authors should explain this point and this limitation.

We would like to thank the reviewers again for their insightful comments. We are happy to hear that our revisions addressed all major concerns. In the following section we address (in *italic*) the remaining points raised by Reviewer 1 and Reviewer 3 and explain the corresponding changes in the manuscript.

Reviewer #1 (remarks to the author)

Generally, the authors addressed our concerns/comments and improved their manuscript. Although I appreciate the addition of Supp fig 2 to better explain the protein engineering, the graphs within that figure are confusing. They have no units on the y-axis, there seem to be many fewer bars than the number of clones analyzed, and the top two graphs are exactly the same even though these should represent different libraries.

Reply: We thank the reviewer for pointing this out. We agree that the graphs (which were thought as cartoon representations for our workflow) were confusing and might led to the impression that they reflect actual measurements. We have now exchanged the barplot with different images to illustrate our workflow, please see Supp.Fig.2.

Supplementary figure S2: Library design. In total 224 mutants were expressed and analyzed in HEK cells. In the first round of mutagenesis, in which we followed the design strategies of Patriarichi et al. 2018, 58 mutants were designed according to the replacement strategy used for the 5-HT2A in Patriarichi et al. 2018 and 12 additional mutants were generated based on dLight. Out of the 58 replacement mutants only 41 showed membrane fluorescence and functionality. The 12 mutants according to dLight were not different in functionality or expression. No obvious pattern for functionality was evident expect a favor for aromatic amino acids in position 2 and 3 of the C-terminal linker. In a second round of mutations, several strategies were applied in parallel to further increase the number of functional mutants: First, position 4 and 5 in the N-terminal linker 1 were again randomly mutated, while retaining an aromatic amino (F/W) acid in the C-terminal linker at position 2 and 3. 70 of these mutants were randomly chosen and further analyzed. In parallel, we selected the two mutants with an increase in fluorescence and mutated them further in different linker positions. 44 of these were further analyzed but revealed no further improvement in desired properties. In addition, according to Patriarichi et al. 2018, we changed the insertion site of cpGFP and analyzed another 20 mutants, in which the position of the cpGFP was varied. In a third round of mutations we started with the most promising mutant so far (M34) and mutated either position 1, 2, 3, of the N-terminal Linker or position 3, 4, 5 of the C-terminal linker randomly. None of the 20 additional mutants showed any improvement over the original variant (M34). As mutant M34 already showed a high and stable brightness, a large change in signal amplitude upon application of serotonin and superior expression in the membrane, we decided to characterize M34, from now on termed *sDarken*, further.

Reviewer #2 (Remarks to the Author):

The authors have adequately addressed my comments and concerns.

Reply: We thank the reviewer for this positive assessment.

Reviewer #3 (Remarks to the Author):

Many clarifications have been made in the revised manuscript, but I have two important remaining concerns.

1. The concerns about pH sensitivity have not been adequately addressed. The experiments of Supplementary Figure S4 are not correctly designed to reveal pH effects, either on the baseline fluorescence of the sensor or on the dose-response curve (i.e. on the K_{app} for 5-HT). As requested in the original review, the authors should measure the effects of pH on samples that are NOT separately normalized (in the current Supplementary Fig 4C, each sample has already been baseline subtracted separately and normalized to its initial fluorescence at its particular pH). Without this information, it is impossible to know whether small pH changes can mimic the effect of 5HT.

Ideally the fluorescence should be observed during a pH change. Additionally, different doses of 5-HT should be used to establish whether pH changes the apparent affinity of the sensor for the ligand.

Reply: We now included all additional pH measurements as requested by the reviewer. We include measurements of pH effects on samples that are not separately normalized and can show that in the physiological pH range no significant changes in fluorescence of sDarken can be observed (**Supp.4d**). Please see additional movie (**Movie-S4d**) for an impression of fluorescence fluctuations during pH changes. These measurements show that changes in extracellular pH do not affect the fluorescence of sDarken.

As requested, we also measured dose response curves at different pH values and included them in **Supplementary S4e**. This is quite important and shows that affinity and dynamic range is not substantially affected by pH.

d) HEK cells expressing sDarken did not show any significant fluorescence changes (RM one-way-ANOVA) to Ringer with different pH values as indicated ($n=70$ cells, from 14 trials). e) Dose response curve measured in response to different 5-HT concentrations at different pH values (2 trials per concentration and pH value, at least 18 cells per condition). pH 6.8: $K_d=33$ nM, pH7.0: $K_d=50$ nM, pH7.2: $K_d=44$ nM, pH7.4: $K_d=20$ nM). mean \pm SD, n.s. not significant

As mentioned before, cpGFP is known for being sensitive to pH changes, but as with all other GPCR based biosensors, such as PsychLight and GRAB, the cpGFP domain is located on the cytosolic side of the cell: Intracellular changes of pH values are expected to be only marginal during action potential firing (e.g. Pressler et al 1988: pH changes about ± 0.04 during AP) and they are thus not expected to influence the fluorescence of sDarken substantially. Marginal changes in fluorescence caused by pH fluctuations will be negligible. Since nearly all existing biosensors rely on cpGFP as fluorophore the pH sensitivity is a general issue of all these sensors, including all GCaMPs. Although we observe slightly different K_d values for different pH values this will not influence the effectiveness of sDarken to detect serotonin dynamics within the physiological range. sDarken is designed for the detection of serotonin dynamics. It is not intended to be a ratiometric sensor, which could give detailed information about concentrations.

In any case, given that intracellular pH changes are negligible and that the affinity is not affected by changes in extracellular pH, sDarken will be a versatile tool to analyze serotonin dynamics.

We added following sentences in the main manuscript:

*"No significant differences in fluorescence of sDarken were observed to pH changes in the range of pH 6.8 – 7.4. In addition, dose response curves at different pH values show that affinity and dynamic range is not substantially effected by pH changes (**Supplementary Fig. S4**).*

2. The inverse agonist behavior of the WAY compound (shown in the authors' response and in Supplementary Figure S3) is interesting - but it actually impairs the ability to use WAY to investigate baseline levels of 5HT. As shown in the figure, even with zero starting [5HT], the WAY compound increases fluorescence and decreases the "apparent 5HT concentration", and this could be incorrectly interpreted as a non-zero concentration of 5HT before the WAY is added. The authors should explain this point and this limitation.

We agree with the reviewer. Indeed, an increase in fluorescence after adding WAY (without the presence of 5-HT) hinders the usefulness of WAY to investigate baseline levels. In the main text we did not refer to WAY as an agent to investigate 5-HT baseline levels.

REVIEWERS' COMMENTS

Reviewer #1 (Remarks to the Author):

The authors have addressed my concerns, I recommend publication of this work.

Reviewer #3 (Remarks to the Author):

I appreciate the authors providing new data on the pH dependence (supplementary S4d and S4e). I am generally satisfied with the response.

Some clarification should still be made to the figure legend. "DeltaF/F" is itself a normalization. Is the denominator F (or "F0") always determined at a single pH (e.g. 7.2) for ALL of the deltaF/F values reported? (If not, that would still be separate normalization.). This should be specified in the figure legend.

We are happy to hear that our revisions addressed all major concerns. In the following section we address (in italic) the remaining points raised by Reviewer 3 and explain the corresponding changes in the manuscript.

Reviewer #1 (Remarks to the Author):

The authors have addressed my concerns, I recommend publication of this work.

Thank you again for your insightful comments and positive assessment of our manuscript.

Reviewer #3 (Remarks to the Author):

I appreciate the authors providing new data on the pH dependence (supplementary S4d and S4e). I am generally satisfied with the response.

Thank you again for your insightful comments and positive assessment of our manuscript.

Some clarification should still be made to the figure legend. “DeltaF/F” is itself a normalization. Is the denominator F (or “F0”) always determined at a single pH (e.g. 7.2) for ALL of the deltaF/F values reported? (If not, that would still be separate normalization.). This should be specified in the figure legend.

Reply: In figure S4d fluorescence changes were measured in one consecutive recording with a randomized order of pH values. Normalization ($\Delta f/f$) was performed with the first frame as F0. In figure S4e we performed a separate normalization for each pH value.

We added this clarification in the figure legend of S4.

d) HEK cells expressing *sDarken* did not show any significant fluorescence changes (RM one-way-ANOVA) to Ringer with different pH values as indicated (n= 70 cells, from 14 trials). Fluorescence changes were measured in one consecutive recording with a randomized order of pH values. Normalization ($\Delta F/F$) was performed with the first frame as F0. e) Dose response curve measured in response to different 5-HT concentrations at different pH values (2 trials per concentration and pH value, at least 18 cells per condition). $\Delta F/F$ was separately calculated for each pH value. pH 6.8: Kd=33 nM, pH7.0: Kd= 50 nM, pH7.2: Kd = 44 nM, pH7.4: Kd = 20 nM). mean \pm SD, n.s. not significant.